



# Conceptual Model to Simulate Long-term Soil Organic Carbon and Ground Ice Budget with Permafrost and Ice Sheets (SOC-ICE-v1.0)

Kazuyuki Saito[1], Hirokazu Machiya[1], Go Iwahana[2], Tokuta Yokohata[3], Hiroshi Ohno[4]

[1]Research Center for Environmental Modeling and Application, JAMSTEC, Yokohama, 236-0001, Japan
[2]International Arctic Research Center, University of Alaska Fairbanks, Fairbanks, 99775, USA
[3]National Institute for Environment Studies, Tsukuba, 305-0053, Japan
[4]Kitami Institute of Technology, Kitami, 090-8507, Japan

*Correspondence to*: Kazuyuki Saito (ksaito@jamstec.go.jp)

**Abstract.** Degradation of permafrost is a large source of uncertainty in understanding the behaviour of Earth's climate
system and in projecting future impacts of climate change. In assessing and projecting the relative risks and impacts of
permafrost degradation, the spatial distribution of soil organic carbon (SOC) and ground ice (ICE) provides essential
information. However, uncertainties in geographical distribution and in the estimated range of total amount of stored carbon
and ice are still large. A conceptual and a numerical soil organic carbon–ground ice budget model, SOC-ICE-v1.0, was
developed, which considers essential aspects of carbon and hydrological processes for above ground and subsurface
environments and frozen ground (permafrost) and land cover changes (ice sheets and coastlines), to calculate long-term
evolution of soil organic carbon (SOC) and ground ice (ICE). The model was integrated for the last 125 thousand years, from
the Last Interglacial until today for areas north of 50°N, to simulate the balance between accumulation and dissipation of
carbon and ice. Model performance was compared with observation-based data and evaluated to assess allogenic (external)
impacts on soil carbon dynamics in the circum-Arctic region on a glacial-interglacial time scale. Despite the limitation of
forcing climate data being constructed on the basis of a single Greenland ice core dataset, the simulated results successfully
reproduced temporal changes in northern SOC and ICE, consistent with current knowledge. The simulation also captured
regional differences in different geographical and climatic characteristics within the circum-Arctic region. The model
quantitatively demonstrated allogenic controls on soil carbon evolution by climatological and topo-geographical factors. The
resulting circum-Arctic set of simulated time series can be compiled to produce snapshot maps of SOC and ICE distributions
for past and present assessments or future projection simulations. Despite a simple modelling framework, SOC-ICE-v1.0
provided substantial information on the temporal evolution and spatial distribution of circum-Arctic soil carbon and ground
ice. Model improvements in terms of forcing climate data, improvement of soil carbon dynamics, and choice of initial values
are, however, required for future research.





# 1 Introduction

Degradation of permafrost is a large source of uncertainty in understanding the behaviour of Earth's climate system and in projecting future impacts of climate change (AMAP (SWIPA) 2011, 2017, IPCC, 2013). Understanding the additional loss of soil carbon and release of greenhouse gases (GHGs) induced by permafrost degradation is important. This is because the impact is experienced far outside the cryosphere (Plaza et al. 2019) and this phenomenon may accelerate global warming through positive feedback (Schuur et al. 2011, Schaefer et al. 2014, Dean et al. 2018a).

Three pathways can be considered for additional GHG release to the atmosphere from a warming permafrost. The first occurs in wide areas through slow and mostly reversible warming and deepening of the active layer (upper soil layer that thaws and freezes seasonally) with a longer thawing period, primarily induced by thermal conduction. This pathway has already been well-recognized and widely examined using Earth System Models (ESMs) and Global Climate Models (GCMs) (Koven et al. 2015). The second and third pathways are related to degradation of ice-rich-permafrost. Ice-rich permafrost

(often called *Yedoma*) is found predominantly in Siberia and Alaska (Kanevskiy et al. 2011, 2013, Murton et al. 2015, Jorgenson et al. 2015, Strauss et al. 2016) and contains massive ground ice (60% to 90% by volume) and carbon-rich sediments. Ice wedge formation is the process responsible for producing the huge amount of buried ice, over a long time in a very cold environment (French 2007, Kanevskiy et al. 2011, 2014, Murton et al. 2015, Jorgenson et al. 2015, Strauss et al. 2016). Soil organic content also accumulates alongside over a long time. Once the ground ice melts by triggers such as

lateral erosion on coasts and riversides or wildfires, GHGs trapped in ice are readily and directly released to the atmosphere (Brouchkov and Fukuda 2002). This constitutes the second pathway (direct release). In addition, old immobile soil organic carbon (SOC), stored frozen in permafrost, is exposed to the surface for decomposition, producing new GHGs. This 'secondary release by ice-rich permafrost degradation' constitutes the third pathway (Strauss et al. 2016, Walter-Anthony et al. 2018, Plaza et al. 2019, Turetsky et al. 2020). Depending on the environment where decomposition occurs (i.e., dry and

aerobic, or wet and anaerobic), the resulting gas differs. Carbon dioxide ($CO_2$) is produced mostly in the former case, while methane ($CH_4$) is more likely produced in the latter case (Schuur et al. 2011, Dean et al. 2018b). The aerial extent of ice-rich permafrost is limited but its impact can be global (Murton et al. 2015, Strauss et al. 2016, Turetsky et al. 2020).

In assessing and projecting the relative risks and impacts of permafrost degradation among the three pathways, the spatial

distribution of SOC and ground ice (ICE) provides essential information. The target area is the circum-Arctic because of the high areal occupancy of permafrost and accumulation of SOC and ICE (Murton et al. 2015, Strauss et al. 2016, Brown et al. 1997). The amount of carbon accumulated in northern soils, including peatlands, accounts for a substantial part of the global soil carbon budget (Gorham 19991, Yu et al. 2010, Hugelius et al. 2014, Nichols and Peteet 2019). Currently available maps and data for near-surface SOC (Hugelius et al. 2014, Olefeldt et al. 2016) and ICE content (Brown et al. 1997) in the circum-

Arctic region are compiled from contemporary samples or cores through interpolation or extrapolation using other topo-geographical and geological information. However, the number of samples and cores is often limited or spatially biased due





mostly to the remote and harsh environment (e.g., cold, altitude, lack of access). Thus, uncertainties in geographical distribution (Hugelius et al. 2014) and in the estimated range of the total amount of stored carbon are still large (Gorham 19991, Yu et al. 2010, Nichols and Peteet 2019).

## 1.1 Carbon and ice accumulation model

In our research, we adopted a different approach to estimate the spatial distribution and amount of SOC and ICE. We developed a conceptual numerical model of two boxes (one for above ground and one for the subsurface) to compute the evolution of SOC and ICE, at a timescale long enough to reproduce present-day conditions by covering more than one cycle of development (i.e., initiation, formation, development, maturation, and decay).

Northern soil carbon (mostly but not entirely in peatlands) was formed and developed during the postglacial period after the Last Glacial Maximum (LGM, around 21 thousand years before present, or 21 ka) (Smith et al. 2004, McDonald et al. 2006, Yu et al. 2009, 2010, Beilman et al. 2009, Klein et al. 2013, Xing et al. 2015, Charman et al. 2015, Loisel et al. 2017, Morris et al. 2018, Nichols and Peteet 2019). Soil carbon dynamics are determined by the balance between inputs (how much carbon is deposited and enters into the soil) and outputs (how much carbon is lost by decomposition or transfer from the soil) of carbon in soil layers, and are controlled by autogenic (internal conditions specific to the ecosystems) and allogenic (external conditions such as climate, hydrology, and topography) factors (Belyea and Baird 2006, Lund et al. 2010, Klein et al. 2013, Charman et al. 2015, Loisel et al. 2017, Jassey and Signarbieux 2019). Owing to temporal changes and geographic characteristics in these factors, carbon accumulation profiles, such as initiation periods or basal ages of accumulations (Smith et al. 2004, McDonald et al. 2006, Yu et al. 2009) and accumulation rates (Harden et al. 1992; Nichols and Peteet 2019) differ from region to region; some of the regions studied previously include Siberia (Smith et al. 2004; Beilman et al. 2009), Alaska (Klein et al. 2013), Northeast China (Xing et al. 2015), Canada (Charman et al. 2015), the circum-Arctic region (Yu et al. 2009, 2010, Hugelius et al. 2014, Olefeldt et al. 2016, Nichols and Peteet 2019), the Southern Hemisphere (Patagonia: Loisel and Yu 2013), and around the globe (Morris et al. 2018). Many researchers have modelled soil carbon dynamics with varying complexities and for various targets at different spatial and temporal scales (Jenny et al. 1949, Ingram 1978, Clymo 1984, Harden et al. 1992, Yu et al. 2003, Belyea and Baird 2006, Morris et al. 2018). In this study, we employed a simple conceptual setting to evaluate long-term evolution of generic soil carbon. The analysis was not necessarily limited to peatlands; further, it was based on Clymo-type growth modelling (Clymo 1984, 1992), in which slow carbon processes occurring in the 'catotelm' (the layer underlying the upper 'acrotelm' of faster carbon processes) were implemented (Clymo 1984, 1992, Yu et al. 2003, Belyea and Baird 2006). We also incorporated a key parameter that represents temporal and spatial variations in climatic and topo-geographic conditions (e.g., curvatures, specific catchment areas, continentality, geomorphology, landscape, and fluvial conditions) to evaluate impacts on soil carbon evolution induced by these external factors.





As for hydrology, we adopted a one-box budget for liquid and solid water dynamics, which is much simpler than in land surface models employed in coupled system models (e.g., Rodell et al. 2004) but more flexible to handle ground ice storage. Soil column layering of the conventional land surface scheme with fixed thickness does not properly represent such massive ground ice as in *Yedoma*. Since most of the currently observable active periglacial features of permafrost-related processes, especially massive ground ice in ice-rich permafrost, formed between the Last Glacial Period and the Holocene (Lunardini

1995, French 2007, Kanevskiy et al. 2011, 2013, Murton et al. 2015, Willeit and Ganopolski 2015, Strauss et al. 2016), the integration period was determined to cover the last 125 thousand years since the Last Interglacial (Kukla et al. 2002), which sufficiently covers the carbon accumulation cycle (Morris et al. 2018, Loisel et al. 2017, Yu 2011, McDonald et al. 2006).

    Climate data for the circum-Arctic region (north of 50°N) were reconstructed for the integration period to force the model.

The resulting model outputs can be compiled to produce contemporary spatial maps of estimated SOC and ICE storage for any time slice, including the present day. Such a model is also expected to quantitatively demonstrate the long-term subsurface dynamics of carbon and ice under varying climatic and environmental conditions, reproduce the long-time evolution of carbon and ice accumulation and decomposition (or dissipation), and provide new insights into understanding the effect of external factors on respective dynamics.


    The development of the model and the data used for determining model parameters or driving the model are described in Section 2. Section 3 presents the results, followed by discussion and future research ideas in Section 4.

## 2 Methods

    The developed conceptual numerical model describes essential aspects of subsurface carbon dynamics and hydrological

processes, to calculate the balance of SOC and ICE. The model simulates SOC and ICE accumulation (or dissipation) history in the circum-Arctic region on a glacial-interglacial time scale. Implemented carbon dynamics processes include the supply of carbon from the surface and loss by decomposition under the ground. Implemented hydrological processes include net infiltration (i.e., precipitation minus evapotranspiration and surface runoff) from the surface, base runoff, and phase changes between solid and liquid states. The model is forced annually by climate variables, namely temperature and precipitation.

Major parameters used for carbon and hydrological processes were determined or parameterized using climatic datasets and geographical information described in Section 2.1. In this study, driving data were reconstructed from Greenland ice core data and applied to the circum-Arctic region to calculate evolution of SOC, soil moisture, and ICE.

### 2.1 Datasets used to develop the model

    The datasets used to determine the parameters of the model included several reanalysis data for the historical period (since

1850), simulation outputs from global-scale climate models for preceding periods, specifically from the Climate Model



Intercomparison Project: Phase 5 (CMIP5, Taylor et al. 2012) and the Paleoclimate Model Intercomparison Project: Initiative 3 (PMIP3, Braconnot et al. 2012), and ice core data from the Greenland Ice Core Project (GRIP, Johnsen et al. 1992, 1997).

Long-term air temperature and precipitation data were taken from the SeaRISE project (Sea-level Response to Ice Sheet Evolution; http://websrv.cs.umt.edu/isis/index.php/SeaRISE_Assessment, Bindshadler et al. 2013), which provided a baseline for climatic changes over the last 125 thousand years from the Last Interglacial to the present day (the latter date set as 1950). This dataset was chosen because it was the only gapless time series for the targeted 125 thousand-year period available at the time of model integration. In order to determine detailed changes for more recent years (i.e., after year 850),
we incorporated simulation results from the PMIP3, especially from past millennium runs for the years 850 to 1850 (Braconnot et al. 2012) and historical runs after the year 1850 (Taylor et al. 2012). For data after the year 1900, we used reanalysis data, the University of Delaware reconstruction product (*UDel_AirT_Precip.* Willmott and Matsuura 2001), and ERA-Interim reanalysis data (Dee et al. 2011).

The Global Land Data Assimilation System (GLDAS, Rodell et al. 2004) was used to determine hydrologic and soil parameters. After preliminary analysis of the hydrological outputs of the four models, we selected the Mosaic model outputs because these showed the best settings and results for regions north of 50°N (Rodell and Beaudoing, https://disc.gsfc.nasa.gov/datasets/GLDAS_MOS10SUBP_3H_001/summary, accessed January 23, 2020). The soil type at each grid point was determined from the 'basic soils information' given in the GLDAS dataset. The original 13 categories of
soil type classes (other than 'water', 'bedrock', and 'other') were aggregated in the model into three major classes, i.e., sand, silt, and clay.

Two ecological data sources were used for carbon input parameterization. The first is the result of stage 2 of the GRENE-TEA model intercomparison project (GTMIP, Miyazaki et al. 2015), which compares the performance of several
biogeochemical models regarding the ecosystem carbon budget for the period 1850–2100. The second is the observational dataset compiled from tropical to sub-Arctic Asian sites, "The compilation dataset of ecosystem functions in Asia (version 1.2)" (personal communication, TM Saitoh, Gifu University).

Construction of driving and boundary conditions for integration was based on these data, as described in section 2.3.

**2.2 Model description**

The model SOC-ICE-v1.0 consists of two boxes: the 'above-ground' box and the 'subsurface' box (Fig. 1). The above-ground box is driven by mean annual air temperature (MAAT) and annual total precipitation (Precip), and has attributes such as latitude, longitude, altitude, distance from the closest ocean body, presence or absence of ice sheet cover, and background





carbon dioxide concentration. The model diagnoses seasonality and frozen ground state and calculates the amount of carbon
supply to the subsurface box. The subsurface box updates SOC, soil moisture, and ICE quantities, according to inputs passed
from and climatic conditions determined by the above-surface box. The model was coded in Interactive Data Language(IDL,
Harris Geospatial Solutions, Inc.). Sample model codes and associated data are provided as supplementary materials.

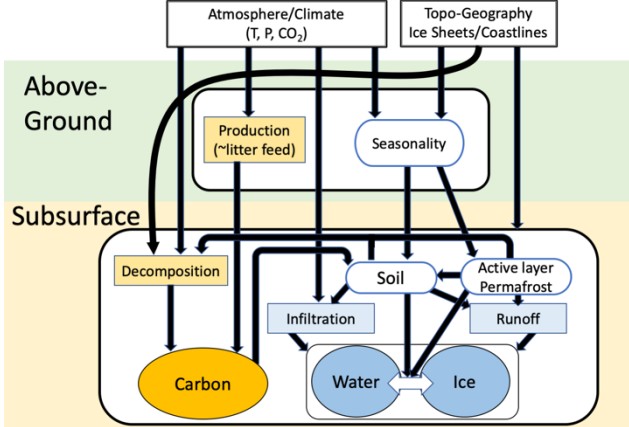

**Figure 1: Schematic diagram of the conceptual numeric model SOC-ICE-v1.0 to calculate soil organic carbon (SOC) and ground ice (ICE) budget.**

### 2.2.1 Above-ground processes

The above-ground box calculates 1) seasonality from local annual mean temperature and its location information
(continentality), and 2) the amount of carbon supply, as litter fall, to the subsurface box.

**Seasonality and presence of frozen ground**

Since the reconstructed temperature data from the SeaRISE project, $T_a$, are on an annual basis, they do not resolve the issue
of seasonality, which is important in inferring subsurface thermal state (i.e., presence of permafrost, seasonal freezing, or no
freezing of ground) (Saito et al. 2014, 2016, Harris 1981). We derived simple relationships between $T_a$ and seasonal
amplitude $T_{amp}$ as a set of functions of location (longitude, latitude), closest ocean, distance from the coast of the closest
ocean, and background climatic state, i.e., glacial (cold) or interglacial (warm) (Supplementary Fig. 1). For the warm period
(defined in this study as 125 ka to 100 ka, and 14 ka to the present day), the present-day climate was assumed to provide
typical variations, and monthly data taken from the ERA-Interim reanalysis were used. In contrast, monthly data taken from
the six PMIP3 models for the LGM simulations were used to derive average climatology for the cold periods (100 ka to 14
ka). The six models were selected so that they provide simulation results for both the LGM and the Holocene Climate
Optimum (mid-Holocene run, 6 ka): CCSM4 (Gent et al., 2011), CNRM-CM5 (Voldoire et al. 2011), IPSL-CM5A-LR





(Dufresne et al. 2013), MIROC-ESM (Watanabe et al., 2011), MPI-ESM-P (Brovkin et al., 2013), and MRI-CGCM3 (Yukimoto et al., 2012). Since the horizontal resolution and the sea/land mask differed between models, coastlines were determined for each model from altitude data. Assuming the sinusoidal seasonal changes in temperature $T_a + T_{amp} \sin t$, the

freezing and thawing indices (FDD and TDD, respectively) were calculated as the cumulative degree day of the temperature below and above 0 °C. The type of underlying frozen ground was then identified based on the classification method developed by Saito et al. (2014, 2016) as climate-driven permafrost (CP, corresponding to continuous permafrost), environmentally-conditional permafrost (EP, corresponding to discontinuous permafrost), long-lasting seasonally frozen ground (Sf), intermittently frozen ground (If, frozen for a short duration, i.e., less than two weeks), or not frozen (Nf).

**Litter fall**

The amount of carbon supply to the subsurface box, calculated as litter fall (in kgC m$^{-2}$ a$^{-1}$), was determined by the combination of MAAT $T_a$ and Precip $P_r$ (Figure 3a.). For simplicity, we did not incorporate carbon type differences inherent in plant functional types (De Deyn et al. 2008). The shape of the litter fall function was determined by fitting the outputs of the biogeochemical models taken from the GTMIP stage 2 project (Fig. 2a–d), i.e., VISIT (Ito 2019), B-BGC (Thornton et al.

2002), SEIB-N (Sato et al. 2016), and CHANGE (Park et al. 2011). Relatively small values of SEIB-N, likely because of its biomass growth still being underway for this integration (cf. Figure S3 in Pugh et al. 2020), did not change the resulting shape significantly. The derived function was then adjusted by multiplying by constant value to the best-fit (in terms of least square errors), to the observational data 'The compilation dataset of ecosystem functions in Asia (version 1.2)' (personal communication, TM Saitoh, Gifu University). The resulting equation was formulated as follows:

$$LitterFall(T_a, P_r, CO_2) = a_1(co_2) \, exp\left(-\frac{(T_a - T_0)^{c_T}}{b_T}\right) \cdot log\left(\frac{P_r + P_0}{b_P}\right), \tag{1}$$

where $T_a(t)$ and $P_r(t)$ denote the values of MAAT and Precip, respectively, of the location in year, $t$. $T_0$ and $P_0$ are the respective baseline values. $a_1$, $b_T$, $c_T$, and $b_P$ are shape parameters. These parameters were determined by curve-fitting to the model outputs (Fig. 2a–d) and observations; $a_1$ and $b_T$ vary depending on the background atmospheric concentration of carbon dioxide $CO_2$ [ppm]. The values of these parameters and baseline values used in this study are summarised in Table 1.

An example of the litter fall distribution under present-day climate conditions is shown in Fig. 2e.




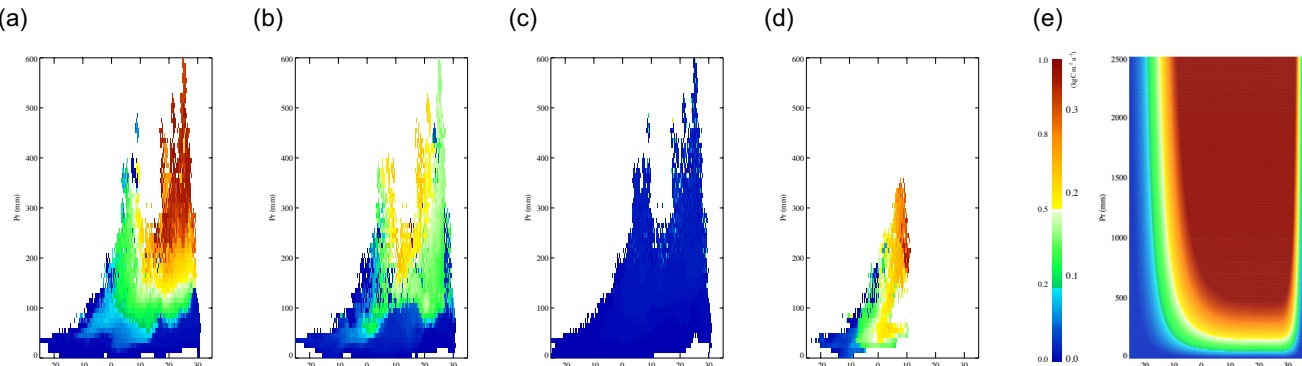

**Figure 2. Distribution of litter fall, the carbon input to the "sub-surface" box, for given mean annual air temperature and annual total precipitation pairs. a) Distribution of litter fall outputs calculated by biogeochemical models and normalized to the respective maximum value: a) VISIT, b) Biome-BGC, c) SEIB-N, and d) CHANGE for global land areas, except for CHANGE, the result of which covered only land areas north of 50°N. The scale is shown on the left-hand side of the colour bar. e) Modelled litter fall distribution based on simulated outputs for the functional shape and calibrated for absolute value by observation data. Note the vertical ranges are different for e). The scale is shown on the right-hand side of the colour bar.**

### 2.2.2 Subsurface processes

The subsurface box has four major functions, 1) SOC budget, 2) water budget in the liquid phase, 3) assessment of freezing

and thawing depths, and 4) phase change between ice and water. The box is not resolved for depth with explicit layering but an active area is assumed to be 3 m deep for hydrological calculations.

### SOC budget

Accumulation and decay of organic carbon in the subsurface box is expressed as a difference equation, adopting Clymo's (1984) peat accumulation model for the catotelm. The change in SOC, $SOC_n$ in the $n^{th}$ year is formulated as Eq. (2).

$$\frac{\Delta SOC_n}{\Delta t} = LitterFall(T_a, P, CO_2) - \kappa_n \cdot SOC_{n-1}, \tag{2}$$

where $LitterFall$ [kgC m$^{-2}$ a$^{-1}$] is the amount of organic carbon deposited on the ground as calculated using Eq. (1), and $\kappa$ is the decomposition rate of $SOC$ [a$^{-1}$], determined at each step by the following relaxation method:

$$\kappa_n = \kappa_{n-1} + \frac{(\kappa_{crit,n} - \kappa_{n-1})}{\tau} \cdot \Delta t . \tag{3}$$

The critical equilibrium rate of decomposition, $\kappa_{crit,n}$, defined as $SOC_n / LitterFall_n$, demonstrates that carbon supply from

the above-ground box and the output (subsurface decomposition) of organic carbon are in balance under the given climate condition. Eq. (3) suggests that the instant decomposition rate approaches the critical equilibrium value on the time scale defined by $\tau$. In this study, we set the values of $\tau$ in a geometric series at 4, 20, 100, and 500 yr to examine climate and topo-geographic controls on soil carbon evolution, after following considerations. Although closely related, $\tau$ does not represent



the soil turnover (e.g., from tens to thousands of years; Perruchoud et al. 1999, Conant et al. 2011, Luo et al. 2019) or
ecological secondary succession (e.g., from some to tens of years after wildfire in permafrost areas; Yohikawa et al. 2003
and Narita et al. 2015, or from tens to hundreds of years under temperate conditions; Svenning and Sandel 2013). It is a
hypothetical variable representing the time scale for decomposition to approach to its equilibrium value under the given
climate and topo-geographic condition. For peatlands in the taiga and tundra that lie between subarctic and arctic regions, it
may take longer than a thousand years to reach an equilibrium due to slow plant growth and carbon decomposition because
of the cool and/or dry environment, while it can happen in a yearly order, $O(1\ year)$ in warm and moist tropical rainforests
where the fast cycle of vegetation growth/death and decomposition facilitates rapid changes (Harden et al. 1992; Vitt et al.
2000a). We also incorporated impacts of permafrost presence and wetness of the ground on SOC dynamics by specifying a
larger value of $\tau$, determined according to frozen and/or wet conditions (e.g., 2500 yr for saturated frozen ground in
continuous permafrost zones and 1500 yr for saturated frozen ground in discontinuous permafrost zones).


**Table 1. List of parameters used in the model**

| Eq. | Category | | | Remarks |
|---|---|---|---|---|
| (1) | Litter fall | $a_1$ | $1.2\left\{1.-\left(0.808-\dfrac{CO2}{1000.}\right)^3\right\}$ | |
| | | $T_0$ | 25.0 | baseline Ta [°C] |
| | | $b_T$ | $10.^\wedge\left(2.14+3.10\dfrac{CO2}{1000.}\right)$, if $T_a > T_0$ <br> $4.5\times10^5$      otherwise | |
| | | $c_T$ | 4.0 | |
| | | $b_P$ | 5.0 | |
| | | $P_0$ | 160.0 | baseline Pr [mm] |
| (4) | Hydrology | $\gamma$ | $0.61 \sim 0.99$ | infiltration ratio [-], depends on soil type, temperature, and frozen state |
| | | $\xi$ | 0.09 (sand), 0.045 (silt), 0.09 (clay) | base runoff ratio [-] |
| (7) | Thermal conductivity | $k^{peat}$ | 0.01 | for peat [W m$^{-1}$ K$^{-1}$] |
| | | $k^{mnl}$ | 1.2 | for mineral soil [W m$^{-1}$ K$^{-1}$] |
| | | $k^{water}$ | 0.6 | for water [W m$^{-1}$ K$^{-1}$] |
| | | $k^{pice}$ | 2.2 | for ice [W m$^{-1}$ K$^{-1}$] |
| (8) | Soil column | $h_b$ | 3000. | soil column depth [mm] |
| | | $\sigma$ | 0.55 (sand), 0.50 (silt), 0.45 (clay) | porosity [-] |


**Hydrological process**

Figure 3 shows a schematic diagram of the subsurface hydrological model for exposed land (i.e., not covered by ice sheet or water). The budget of the liquid-phase water $W_n$ is controlled by Eq. (4) as the balance between the input (the first term on the righthand side) and the output (the second and third terms).

$$\frac{\Delta W_n}{\Delta t} = \gamma P - \xi W_{n-1} - \varphi_n. \tag{4}$$

The first term in Eq. (4) refers to annual net precipitation (i.e., precipitation – evapotranspiration – surface runoff) with $\gamma$ denoting the ratio of subsurface infiltration to total precipitation. The second term refers to base runoff as a function of water storage in the liquid phase, and $\xi$ is a parameter for the ratio of base runoff. The third term refers to new ice freezing or melting at a time step. Soil moisture [mm] in the active area overflows as runoff when it exceeds the saturation soil moisture ($h_a = \sigma h_b$), where $\sigma$ is porosity, and $h_b$ is the depth of the active area set to 3000 mm in this study (Fig. 3). In contrast, the overall storage of ICE at location $I_n$ has no limitation (to mimic the development of ice wedge) and is updated using Eq. (5). The computation of $\varphi$ is described in the next subsection.

$$I_n = \varphi_n + I_{n-1} \tag{5}$$

The parameters of soil characteristics (porosity $\sigma$, infiltration rate $\gamma$, and base runoff ratio $\xi$ of the area) are summarized in Table 1.

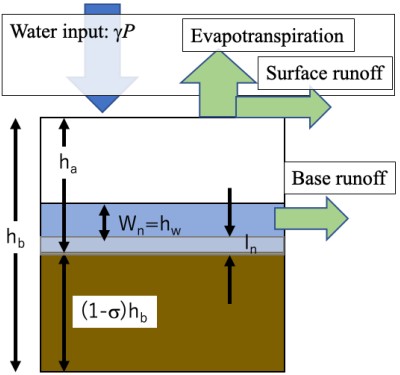

**Figure 3. Schematic diagram for hydrological processes of the subsurface box.**

**Assessment of freezing and thawing depths**

The changes in amount of ICE, $\varphi_n$, are analysed through the energy balance between ground freezing and thawing processes. They are proportional to the depth of freezing $d_n^f$ and thawing $d_n^t$, which are empirically determined by Eq. (6a–b) from





ground thermal conductivity (for the frozen state, $k_n^f$, and thawed state, $k_n^t$, respectively) and respective freezing and thawing indices calculated in the above-ground box (i.e., FDD and TDD).

$$d_n^f = \sqrt{\frac{2k_n^t \cdot FDD}{W_n/_{h_b} \rho_w \cdot \lambda}} = \alpha_f \sqrt{FDD}, \tag{6a}$$

$$d_n^t = \sqrt{\frac{2k_n^f \cdot TDD}{W_n/_{h_b} \rho_w \cdot \lambda}} = \alpha_t \sqrt{TDD}, \tag{6b}$$

where $\rho_w$ is the density of water, and $\lambda$ is the latent heat of fusion. Thermal conductivity is evaluated from carbon and water content using Eq. (7).

$$k_n^{t,dry} = \frac{SOC_n}{\chi^{t,dry}} k^{peat} + \left(1 - \frac{SOC_n}{\chi^{t,dry}}\right) k^{mnl}$$

$$k_n^t = \frac{W_n}{\chi^{t,wet}} k^{water} + \left(1 - \frac{W_n}{\chi^{t,wet}}\right) k_n^{t,dry} \tag{7}$$

$$k_n^f = \frac{I_n}{\chi^f} k^{ice} + \left(1 - \frac{I_n}{\chi^f}\right) k_n^t$$

where $k^{peat}, k^{mnl}, k^{water}, k^{ice}$ denote thermal conductivity values for carbon-containing and mineral parts of the soil, water, and ice, respectively. The relative amounts of thawed dry or wet soil, and frozen soil are defined by Eq. (8).

$$\chi^{t,dry}(SOC_n, h_b; \sigma) = (1 - \sigma)h_b + SOC_n$$

$$\chi^{t,wet}(W_n, SOC_n, h_b; \sigma) = (1 - \sigma)h_b + SOC_n + W_n = \chi^{t,dry} + W_n, \tag{8}$$

$$\chi^f(I_n, SOC_n, h_b; \sigma) = (1 - \sigma)h_b + SOC_n + I_n = \chi^{t,dry} + I_n$$

**Phase change between water and ice**

The change in ice amount is calculated as follows:

$$\varphi = \begin{cases} -I_n \\ \tilde{\varphi} \\ W_n \end{cases} \quad \text{if} \quad \tilde{\varphi}: \begin{cases} \cdot < -I_n \\ -I_n \leq \cdot \leq W_n, \\ W_n < \cdot \end{cases} \tag{9}$$

where $\tilde{\varphi}$ is defined by

$$\tilde{\varphi} = \beta \left(d_n^f - d_n^t\right) \frac{W_n}{h_b}. \tag{10}$$



$\beta$ is a parameter to control the distribution of energy in melting or freezing of water, $\beta_{freeze} = 0.5$; $\beta_{thaw} = 1.0$. The parameter values not listed here are summarised in Table 1.

## 2.3 Driving and boundary condition data

The model is driven by inputs of MAAT, Precip, and atmospheric carbon dioxide concentration, along with geographical information: longitude, latitude, continentality, and land condition types (i.e., exposed land, under water, or under ice sheets). In this study, the model was integrated for the last 125 thousand years north of 50° in the Northern Hemisphere with 1-degree resolution, aligning with the grid system of the employed dataset for ice sheet evolution (i.e., ICE-6G_C).

## 2.3.1 Forcing data for 125 thousand years

The baseline 125 thousand-year time series of annual temperature and precipitation was taken from the SeaRISE (Sea-level Response to Ice Sheet Evolution; Bindshadler et al. 2013) project. The time series presents deviations from current mean temperature, or ratios to the current precipitation amount. The SeaRISE time series needs present-day climatology data for a specific location. The present-day climatology was computed from ERA-Interim reanalysis data for the years 1979–2016. Moreover, the SeaRISE time series has low temporal resolution in the recent millennium (i.e., 100-yr intervals). We examined the CMIP5/PMIP3 models for overall goodness of the reproduced time series in the circum-Arctic region and used the IPSL simulation results. The past climate anomaly time series for the period 850–1850 was refined by its 'last millennium' run, and for the period 1850–1900 by its 'historical' run. Similarly, the anomaly time series for the period 1900–2006 was constructed using the University of Delaware reconstruction product (*UDel_AirT_Precip*).





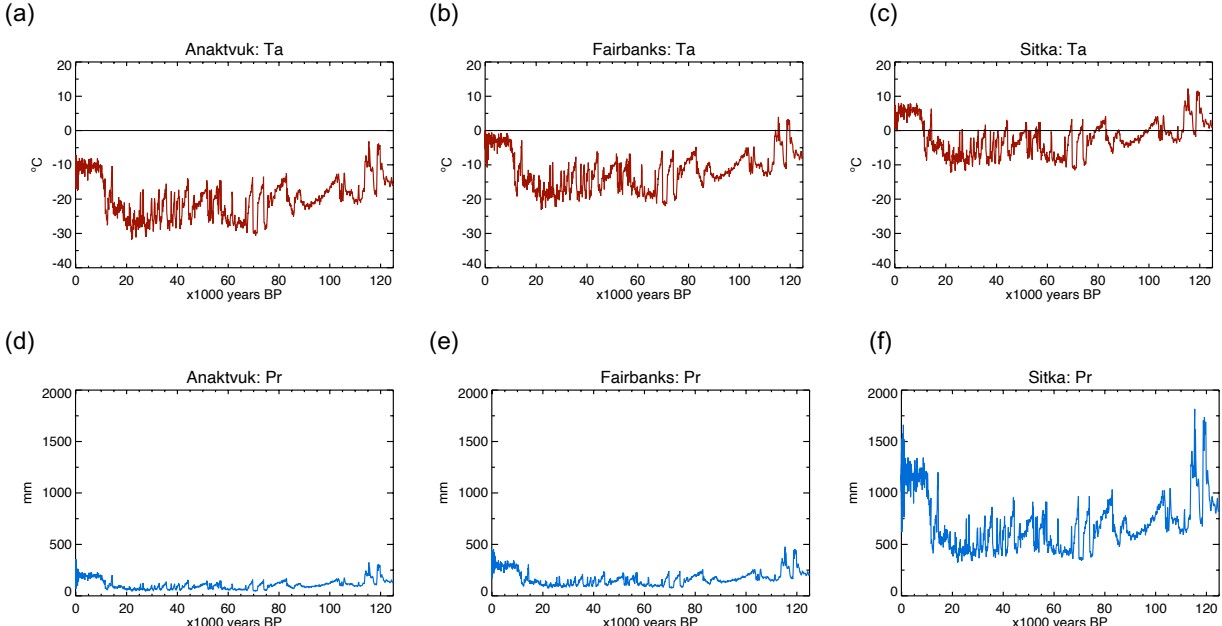

**Figure 4: Example time series of climate forcing data (temperature, precipitation), and diagnosed litter fall. The temporal variations in climate forcing data were reconstructed from Greenland ice-core data (the SeaRISE project), while present-day climatology at the 1-degree grid point was derived and interpolated from ERA-Interim reanalysis. The litter fall time series were computed from Eq. (1). Mean annual air temperature (MAAT; in °C) at the Alaskan grid point closest to a) Anaktuvuk (69.5°N), b) Fairbanks (64.5°N), and c) Sitka (57.5°N). d), e), and f) Same as a), b), and c) except for annual total precipitation (in mm).**

Considering the meridional dependence (i.e., polar amplification effect) of $\delta T_a$, the amplitude of temperature variations between glacial and interglacial periods and the amplitude of the SeaRISE anomaly temperature time series were reduced southward within the 50 and 70 meridional band at a rate of 0.25 °C per 1 degree from its original value, 23.3 °C, which was determined by preliminary analysis on $[\partial(\delta T_a)/\partial\,lat]$ for six PMIP3 models (Supplementary Fig. 2). These time series were then combined with present-day climatology to produce a 125 thousand-year time series for each 1-degree resolution grid

point. Examples of the reconstructed time series of temperature and precipitation for three different locations in Alaska are shown in Fig. 4.

### 2.3.2 Boundary conditions

For the last 125 thousand years of the integration period, there have been substantial changes in the presence and thickness of ice sheets, as well as in altitudes and coastlines, in the circum-Arctic area. These surface boundary conditions exert a large

influence on calculation of subsurface carbon and ice dynamics through various processes such as submergence, uplift, burial under ice sheets, and removal by glacial dynamics. We used the ICE-6G_C datasets (Argus et al. 2014, Peltier et al. 2015) to determine altitude, the areal ratio of land, water, and ice cover, and ice thickness for the original 1-degree grid points. Each grid point has three sections, i.e., exposed land, under ice, and under water (e.g., sea, lake), following the areal





fraction of land and ice of the dataset. The amounts of carbon, ice, and water are reshuffled due to changes in land cover

fractions, in addition to internal dynamics of carbon and water determined by Eqs. 2 and 4. When some portion of the grid becomes ice-free (melting of ice sheet), water from the melting ice is added to the precipitation input (in Eq. 4), and the SOC amount corresponding to the newly exposed areal fraction is lost from the grid's storage to reflect basal ablation. When some portion of the grid submerges, the SOC amount and ice content, as well as frozen ground condition, belonging to this areal portion are kept unchanged but the water content to the portion becomes saturated.

## 3 Results


The amounts of SOC, ICE, and soil moisture were computed using the SOC-ICE-v1.0 model. The calculation was performed on the 1-degree interval grid system. Before full integration, the model was spun up by constant initial forcing and using boundary data for 5000 yr, from the uniform initial values of 25.0 kgC m$^{-2}$ of SOC and 500 mm of soil moisture.

(a)                                                                         (b)

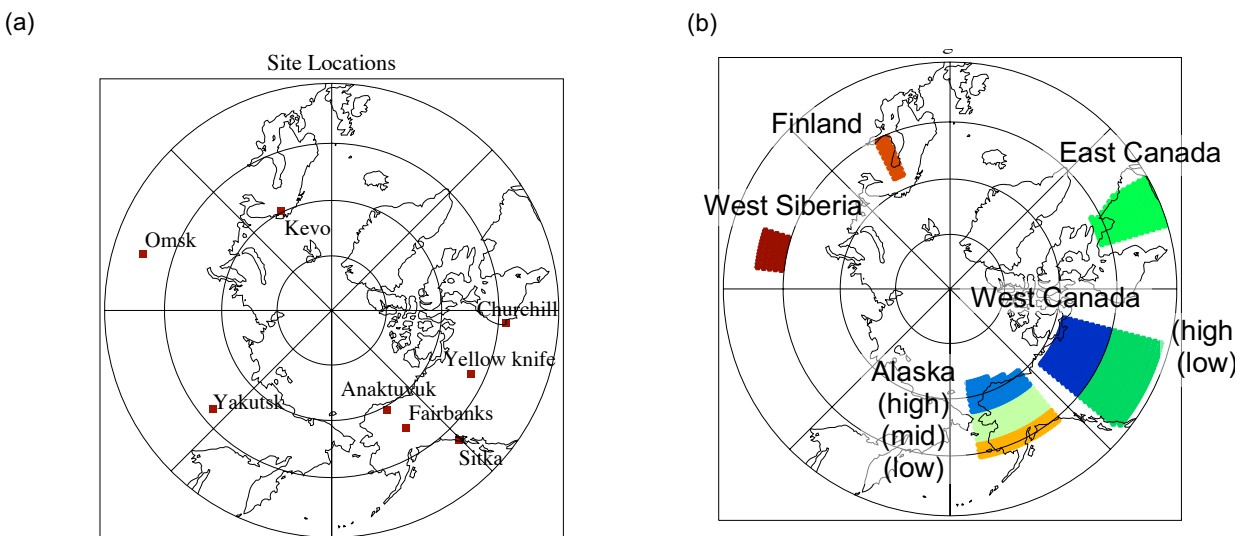

**Figure 5: a) Locations of the eight example sites for comparison of simulated time series. b) Areal extents of the eight sub regions for analysis of the accumulation history of SOC and ICE.**





**Table 2. Comparison of simulated SOC and ICE with observation-based data for eight locations (longitude and latitude denote the nearest 1-degree resolution grid point).**

| Locations | Soil[a] | Var. | Obs-based[b] | $\tau = 4$ | $\tau = 20$ | $\tau = 100$ | $\tau = 500$ |
|---|---|---|---|---|---|---|---|
| Anaktuvuk | silt | SOC[c] | 66.3 | 48.7 | 48.7 | 48.7 | 44.6 |
| (69.5ºN, 150.5ºW) | | ICE[d] | 10-20% | 30.6 | 30.6 | 30.6 | 30.6 |
| Ta: -8.5 ºC, Pr: 175 mm | | SM[e] | 0.56 | 0.92 | 0.92 | 0.92 | 0.92 |
| Fairbanks | silt | SOC | 43.8 | 13.7 | 17.7 | 25.0 | 39.8 |
| (64.5ºN, 147.5ºE) | | ICE | 0-10% | 0.3 | 0.3 | 0.3 | 0.3 |
| Ta: -3.1 ºC, Pr: 246 mm | | SM | 0.54 | 0.78 | 0.78 | 0.78 | 0.78 |
| Sitka | silt | SOC | 0.0 | 1.0 | 3.1 | 13.1 | 30.3 |
| (57.5ºN, 135.5ºW) | | ICE | 0% | 0.0 | 0.0 | 0.0 | 0.0 |
| Ta: 5.3 ºC, Pr: 657 mm | | SM | 0.28 | 0.52 | 0.52 | 0.52 | 0.52 |
| Yellowknife | silt | SOC | 14.1 | 11.7 | 15.0 | 23.7 | 35.6 |
| (62.5ºN, 114.5ºW) | | ICE | 10-20% | 20.3 | 20.3 | 20.3 | 20.2 |
| Ta: -6.9 ºC, Pr: 96 mm | | SM | 0.84 | 0.87 | 0.87 | 0.87 | 0.87 |
| Churchill | silt | SOC | 166.8 | 23.7 | 26.6 | 33.7 | 36.5 |
| (58.5ºN, 94.5ºW) | | ICE | 10-20% | 28.5 | 28.5 | 28.5 | 28.5 |
| Ta: -8.8 ºC, Pr: 224 mm | | SM | 0.73 | 0.49 | 0.49 | 0.49 | 0.49 |
| Kevo | sand | SOC | 60.1 | 0.4 | 4.4 | 17.6 | 38.4 |
| (69.5ºN, 27.5ºE) | | ICE | 0-10% | 0.0 | 0.0 | 0.0 | 0.0 |
| Ta: 0.6 ºC, Pr: 181 mm | | SM | 0.43 | 0.07 | 0.07 | 0.07 | 0.07 |
| Omsk | clay | SOC | -[f] | 6.5 | 9.4 | 13.2 | 34.4 |
| (54.4ºN, 73.5ºE) | | ICE | 0% | 0.0 | 0.0 | 0.0 | 0.0 |
| Ta: 0.4 ºC, Pr: 207 mm | | SM | 0.9 | 0.74 | 0.74 | 0.74 | 0.74 |
| Yakutsk | silt | SOC | 70.2 | 39.0 | 39.0 | 39.0 | 40.9 |
| (62.5ºN, 129.5ºE) | | ICE | 10-20% | 16.2 | 16.2 | 16.2 | 16.2 |
| Ta: -11.5 ºC, Pr: 138 mm | | SM | 0.24 | 0.59 | 0.59 | 0.59 | 0.59 |

a) Soil types are determined from the basic soils information used for the 1-degree resolution Global Land Data Assimilation System (GLDAS. Rodell et al. 2004) dataset (https://ldas.gsfc.nasa.gov/gldas/soils. Accessed on March 10, 2020).

b) Soil organic carbon amount is taken from Olefeldt et al. (2016); Ground ice content category is taken from Brown et al. (1998).

c) Observation-based and simulated Soil organic carbon is in kgC m[-2].

d) Simulated ground ice is in meter.

e) Soil moisture in terms of saturation ratio was calculated respectively from the GLDAS Mosaic product (assimilated data) and the SOC-ICE-v1.0 results, assuming the same porosity.

f) No data was found in the vicinity.





Here, we examined the behaviour of the simulated time series for SOC, ICE, and soil moisture through the glacial and
interglacial periods at the selected eight locations with different climatic characteristics in the circum-Arctic regions. The
locations denote the grid points closest to each specified site shown in Fig. 5a. Of these, three sites are in Alaska: Anaktuvuk
on the North Slope (continuous permafrost, Jones et al. 2009, Hu et al. 2015, Iwahana et al. 2016), Fairbanks in Interior
Alaska (discontinuous permafrost, Miyazaki et al. 2015, Sueyoshi et al. 2016), and Sitka in southeast Alaska on the Pacific
coast (seasonally frozen ground). Sitka is the warmest and most pluvial site of the eight. Two sites are in Canada:
Yellowknife by the Great Slave Lake in the Northwest Territories (discontinuous permafrost) and Churchill in Hudson Bay
Lowlands (continuous permafrost, Dyke and Sladen 2010, Sannel et al. 2011). Both of these were under the Laurentide Ice
Sheet during the Last Glacial period and started carbon accumulation after deglaciation at different times (Dyke 2005). Three
sites are in Eurasia. Kevo in northern Finland (Miyazaki et al. 2015, Sueyoshi et al. 2016), which was covered by the
Fennoscandia Ice Sheet during the Last Glacial period, has oceanic influence and in the discontinuous permafrost zone.
Omsk in southwestern Siberia (seasonally frozen ground) has continental characteristics and has been ice-free. Yakutsk is in
East Siberia (continuous permafrost, Miyazaki et al. 2015, Sueyoshi et al. 2016). Anaktuvuk and Yakutsk are in areas that
include the ice-rich permafrost (*Yedoma*) region (Murton et al. 2015, Kanevsky et al. 2011). Geographical and climate data
of these locations are shown in Table 2.

**3.1 Time series analysis**

Table 2 shows the results of the simulated present-day contents and corresponding observation-based data at the eight
selected locations with respect to SOC, ICE, and soil moisture (relative to the saturation level). In Table 2, the calculated
SOC contents with different values of $\tau$ are compared with total SOC amount compiled by Olefeldt et al. (2016) at the
nearest point. The computed ICE content was compared with ground ice information compiled by Brown et al. (1997),
which is the only currently available distribution data covering the entire circum-Arctic area. Note that Brown et al. (1997)
categorised ICE distribution by volumetric content, i.e., none, 0–10%, 10–20%, or over 20%, depending on the type of
overburden. The larger the value of $\tau$, the larger the simulated SOC amount for the present day. On the contrary, ICE content
showed almost no sensitivity to $\tau$ under current formulae (Eqs. 4–10). The resulting ranges at eight locations and inter-site
variations in the simulated present-day SOC and ICE contents were largely consistent with observation-based data, except
for underestimation of SOC and overestimation of ICE at the Churchill site. Moreover, the value of $\tau$ has a discernible
control over the simulated present-day SOC amount for all locations except Anaktuvuk and Yakutsk, which have been
almost entirely in continuous permafrost zones throughout the integration period (Fig. 6).







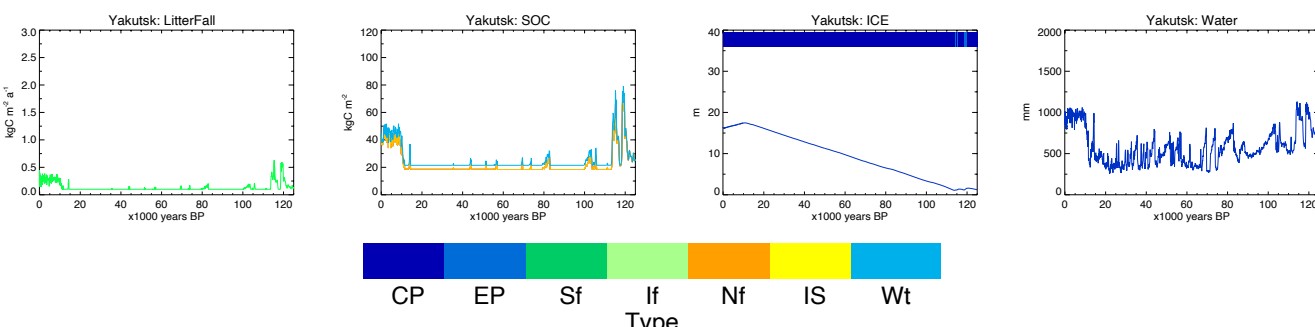

**Figure 6: Results of model simulations for the eight circum-Arctic sites shown in Figure 5. 125-thousand-year time series of litter fall (in kgC m$^{-2}$; leftmost column), soil organic carbon (SOC in kgC m$^{-2}$; second from left), ground ice (in meters; second from right) and ground water in liquid phase (in mm; rightmost column) are shown for different values of τ (red: 4 years, orange: 20 years, yellow: 100 years, blue; 500 years). The legend shows land cover types for continuous permafrost (CP, deep blue), discontinuous permafrost (EP, blue), seasonally freezing ground (Sf, green), intermittently frozen ground (If, pale green), no freezing (Nf, orange), ice sheets (IS, yellow), and water (Wt, pale blue).**


Figure 6 shows the 125-thousand-year timeseries of model outputs for litter fall, SOC, ICE, and soil moisture at eight locations. Litter fall (the leftmost column in Fig. 6) increased in proportion to annual temperature and precipitation (Eq. 1, Figs 2a and 4). Although the values of litter fall appeared to be larger than observed values, the resulting SOC amount and its circum-Arctic distribution did not show overestimation (see the SOC column of Fig. 6; cf. Saito et al. 2020 in review). This

result is discussed in section 4, along with suggestions for future improvements. The simulated SOC changes showed a tendency at all locations for accumulation to be active during warm (i.e., interglacial) periods and inactive during cold (i.e., glacial) periods, consistent with existing knowledge (Vitt et al. 2000a, 2000b, Charman et al. 2015). The accumulation time series shows high sensitivity to τ, except in continuous permafrost zones, in line with the result of dependency of present-day SOC amounts on τ (Table 2). These behaviours demonstrated adequate functionality of τ to represent allogenic controls of

external (i.e., climatic, topographic and/or land composition) conditions over carbon dynamics in terms of the time required to shift to an equilibrium under meandering climate conditions (Loisel et al. 2017, Belyea and Baird 2006).

Accumulation of ICE at some time during the glacial period and its decrease after the onset of deglaciation were observed at all locations except for Sitka, where no ICE was accumulated throughout the period. In continuous permafrost zones with no

ice sheet coverage, i.e., in Anaktuvuk and Yakutsk, ICE steadily accumulated during glacial periods and persists to date, despite some melting in the post-glaciation period after the LGM. For sites entirely covered with land ice during the glacial period, e.g., Yellowknife, Churchill, and Kevo, accumulation of ICE was computed even under land ice. This can be interpreted as the potential amount of buried massive ice at those locations, but may need further consideration and/or modifications in model formulations. Moreover, note that absolute values of simulated ICE do not necessarily show the *in-*

*situ* amount found in the soil layer. Rather, they indicate a relative value to be compared among different locations for contemporary spatial variations or temporal development.



Temporal changes in soil moisture (in liquid phase) are demonstrated in the rightmost column of Fig. 6. In (continuous) permafrost zones, this value should be interpreted as the amount of liquid water available in summer, not the total amount of
liquid water kept unfrozen during winter or over the entire period. Although it is difficult to examine the likeliness of simulated long-term changes and inter-site differences in quantitative comparisons with observation-based evidence due to lack of data, we can still interpret qualitative behaviour. The water level after deglaciation depended largely on precipitation amount and frozen ground type (e.g., continuous, discontinuous permafrost, or seasonally frozen ground). At the Anaktuvuku and Yakutsk sites, which were underlain by cold continuous permafrost, ground was dry during the glacial
period and wet in the warmer Holocene because the length and depth of the thawed layer were limited under glacial conditions, with most liquid water frozen and stored as ice, while the active layer was thicker and persisted longer during the Holocene. In a warmer continental location, like Omsk, the water content was higher when underlain by permafrost than when ice-free, to which the decrease in base runoff and evapotranspiration may likely contribute. In Fairbanks, which is also an interior city but cooler and wetter than Omsk, the average and range of temporal variations remained unchanged for the
entire integration period. Greater availability of water due to a wetter climate may have contributed to larger fluctuations during the Holocene than in the Omsk case. At those locations covered under land ice during the glacial time (e.g., Sitka, Yellowknife, Churchill, and Kevo in Fig. 6), soil moisture was saturated under ice sheets (n.b., by formulation) but commonly became drier once the ice sheets retreated.

## 3.2 Regional analysis for the deglaciation period

We examined the simulated results of carbon dynamics-related characteristics, i.e., basal age distribution and accumulation rates for the post-glacial SOC accumulation. We specified eight circum-Arctic regions and compared results for these with values reported in the literature (Yu et al. 2009, Smith et al. 2004, McDonald et al. 2006, Jones and Yu 2010). The locations and areas of these eight regions are defined as follows: Alaska (high latitudes) (67.5–73.5°N, 169.5–139.5°W), Alaska (middle latitudes) (62.5–67.5°N, 169.5–139.5°W), Alaska (low latitudes) (59.5–62.5°N, 169.5–139.5°W), West Canada
(high latitudes) (60.5–69.5°N, 129.5–103.5°W), West Canada (low latitudes) (51.5–60.5°N, 129.5–103.5°W), East Canada (44.5–62.5°N, 73.5–59.5°W), Finland (60.5–68.5°N, 22.5–27.5°E), and West Siberia (55.5–60.5°N, 72.5–84.5°E) (Fig. 5b).

### 3.2.1 Basal age of carbon accumulation

Figures 7a to 7c show histograms of the basal age of post-glacial soil carbon accumulation over the entire circum-Arctic domain (north of 50°N). The observational studies showed the peak of northern high-latitude SOC initiation after the LGM
at 11–9 ka, despite regional differences (Morris et al. 2018, Jones and Yu 2010, McDonald et al. 2006, Loisel et al. 2017, Yu et al. 2009, 2010; Smith et al. 2004). In comparison, simulated results showed a similar peak of increased initiation during the same period but shifted earlier by a millennium at 12–10 ka. There may be multiple reasons for this discrepancy. One possibility is related to the forcing data. Local climatic history varied from location to location (Morris et al. 2018), while





climate data reconstructed from a Greenland ice core was in phase at all locations (Fig. 4a–c). This is also revealed by the
unequivocal concentration of the initiation peak in the simulated results (n.b., the logarithmic vertical scale in Fig. 7).
Another possibility is insufficiency in the parameterisation of carbon input (Eq. 1) and output (Eqs. 2–3). The discrepancy
may also be attributed to differences or technical limitations in the determination of SOC initiation. For numerical data, the
basal age can be defined as the first timestep with non-zero accumulation of SOC after the LGM. The limit of detection in
the laboratory may simply not work at the same resolution. Despite these limitations, the results capture the impacts of
external changes (i.e., an allogenic control) on carbon dynamics during the deglaciation period.

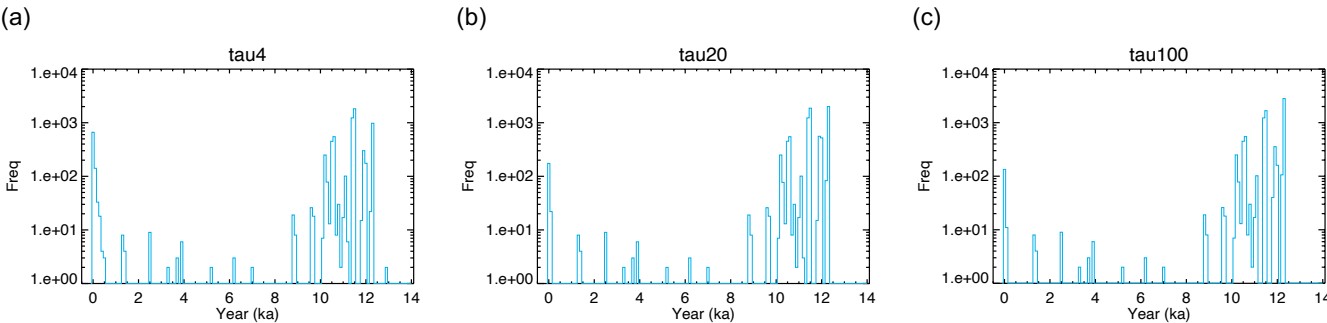

**Figure 7: Histograms of circum-Arctic basal age distribution of the SOC accumulation for different values of $\tau$: a) 4 years, b) 20 years, and c) 100 years.**

### 3.2.2 SOC accumulation rates

Figure 8 shows temporal changes in the post-glacial SOC accumulation rate for eight regions (Fig. 5b). Accumulation rates
were calculated from the original simulated annual time series, aggregated for regions for every millennial interval, and then
sorted to derive percentiles. The lowest and highest whiskers of the box-whisker plots show the 10th and 90th percentiles,
respectively, the lower and upper edges of the box show the 25th and 75th percentiles, respectively, and the coloured bar in
the box shows the 50th percentile (median). The general tendencies were reasonably reproduced such that the accumulation
rate is high at 9–12 ka and in the last millennium, while it is low to modest in between (cf. Plates 3b and 5d in Yu et al. 2008,
Figs. 1c and 3 in Yu 2012). The average accumulation rate is estimated as 18.6 gC m$^{-2}$ a$^{-1}$ for the Northern Hemisphere
extratropical climate in the Holocene (Yu et al. 2009). Simulated accumulation rates for different circum-Arctic sites and
regions in Figure 8 agree with observation-based estimates: 24.1 gC m$^{-2}$ a$^{-1}$ in Fairbanks, Alaska (middle latitude), 5.7–13.1
gC m$^{-2}$ a$^{-1}$ in Alaska (low latitude), 15.6–31.7 gC m$^{-2}$ a$^{-1}$ in West Canada (low latitude), 7.0–30.6 gC m$^{-2}$ a$^{-1}$ in East Canada,
12.9–22.5 gC m$^{-2}$ a$^{-1}$ in Finland, and 21.9–70.6 in West Siberia (Yu et al. 2008).





Simulated accumulation rates can be negative while those estimated from excavated cores can only show positive values. The large negative values found in the 12–13 ka bin are thus worth investigating. Extensive decomposition for this period is consistent with the fact that the basal age derived from the core samples does not go back before 13 ka.

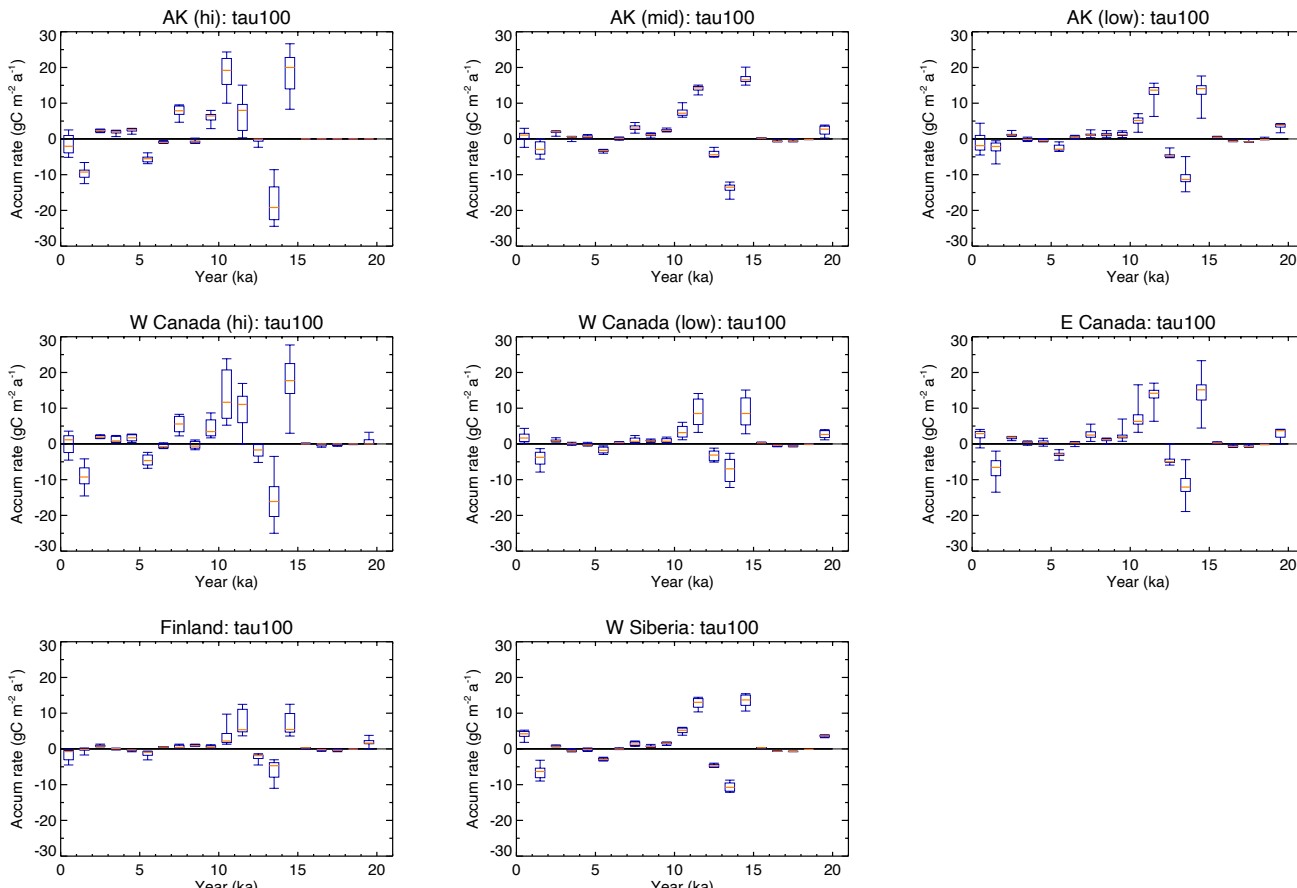

**Figure 8: Millennial changes in the carbon accumulation rate (gC m⁻² a⁻¹) after the LGM for different circum-Arctic regions. The statistical distribution for each millennial period is shown by a box-whisker plot. The lowest and highest whiskers of the box-whisker plots show the 10ᵗʰ and 90ᵗʰ percentiles, respectively. The lower and upper edges of the box show the 25ᵗʰ and 75ᵗʰ percentiles, respectively. The 50ᵗʰ percentile (median) is shown by the coloured bar in the box.**

### 470 3.2.3 Changes in ground ice

Similar plots for temporal changes in the regional budget of ICE after the LGM are shown in Fig. 9. In all eight regions, general accumulation of ground ice was observed until the end of 15 ka. During the 14–15 ka period, large melting of ground ice occurred in relatively warmer areas, i.e., West Canada (low), East Canada, Finland, and West Siberia. The ground ice melted at 9–11 ka but at a lower rate. Ground ice accumulation continued after 12 ka only in colder regions, such as high-





latitude areas in Alaska and West Canada. These regional differences in and characteristics of simulated ICE evolution correspond well to today's conditions, but it is difficult to validate these temporal changes using observation-based sources.

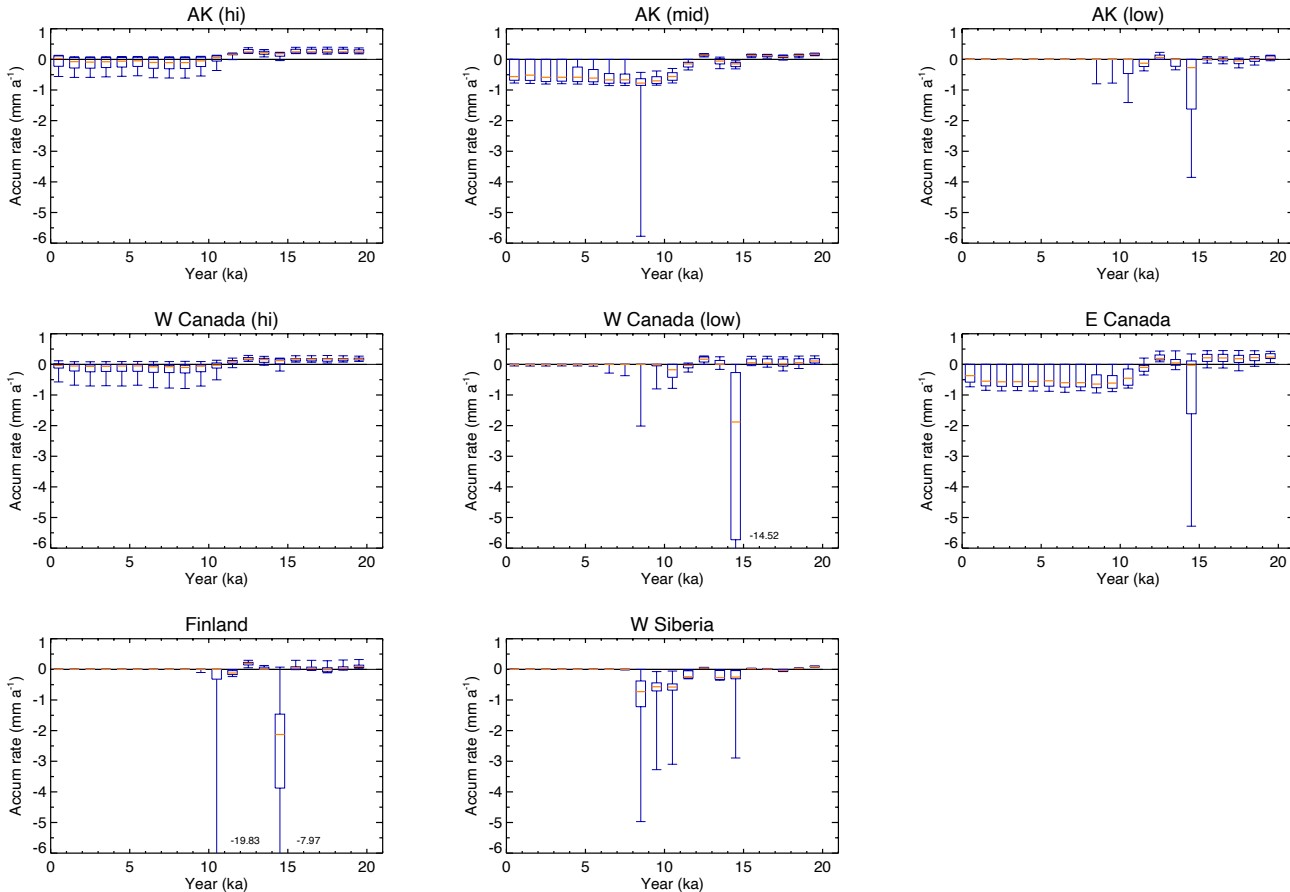

**Figure 9: Same as Figure 8, except for ground ice accumulation in mm a⁻¹.**

# 4 Discussion

## 4.1 Implications

The simulated time series of SOC, ICE, and soil moisture for each location can be compiled to produce snapshot maps for any period in the last 125 thousand years. In fact, the present-day distributions of simulated SOC and ICE were mapped for the area north of 50°N and presented in an accompanying paper (Saito et al. 2020, under review). In the paper under review, we created maps of SOC amounts for each $\tau$ at the original 1-degree resolution and discussed differences in and spatial characteristics of simulated distributions for different values of $\tau$. In addition, we developed a methodology to associate the

value of $\tau$ with local topographic-hydrological features derived from a 2 arc-minute digital relief model and produced



respective high-resolution circum-Arctic maps of SOC and ICE for comparison with currently available observation-based data (i.e., Hugelius et al. 2014, Olefeldt et al. 2016, Brown et al. 1997). Similar maps of SOC, ICE, and soil moisture can be produced for different time points (e.g., the Holocene Optimum and the LGM) to examine the areal development of variables in the region. These snapshot maps can also be used for initial and/or boundary conditions for Earth System Models (ESMs)

or large-scale terrestrial eco-climate models to assess past or present states, or to project future impacts on potential release of GHGs induced by permafrost degradation. Yokohata et al. (2020 in review) partly used the results derived from the present-day snapshot to quantify the relative impacts resulting from the three pathways described in section 1. Such results can provide localised information on the mechanism of permafrost-related GHG releases (e.g., vulnerable areas, potential speed of development) to help stakeholders (i.e., policy- and decision-makers, as well as residents at local to global levels)

adapt to, stabilize, or mitigate climate change consequences.

### 4.2 Future improvements

The simple framework of the modelling concept and the reconstruction methodology of forcing/boundary data creation were useful to reproduce the evolution and to draw a big picture of allogenic control over long-term carbon dynamics. Nevertheless, there is room for further improvement. Below is a list of possible problems with the current version.


The first relates to the forcing data. In order for the resulting reproduction maps to present locality-prone profiles more adequately, the forcing climate data time series need to include information more specific to their history than in this study. Temporal variations in the current method are solely based on a single Greenland ice core. Thence, they are basically in-phase within the simulated circum-Arctic domain, although they incorporate spatial variations such as changes in coastlines,

altitudes, and ice sheets, as well as the meridional gradient on the amplitude of long-term temperature variations (i.e., polar amplifications). However, temporal changes in climate have distinct regional components (Frenzel et al. 1992, Alley et al. 2002, Nakagawa et al. 2003) and behaviours at different time scales (Esper et al. 2002, Cook et al. 2004, Mann et al. 2009). As a result, the initiation timings of the Holocene carbon accumulation were heterogeneous globally (Morris et al. 2018).

The second problem relates to formulation of carbon dynamics (Eqs. 1–3). With regard to the quantity in the carbon budget, the calculated litter fall tended to be overestimated, as shown in Fig. 6. Equation (1) was designed to express the upper envelop of the litter fall for the range of temperatures and precipitation levels considered (i.e., -25 °C to 35 °C and 0 mm a$^{-1}$ to 600 mm a$^{-1}$, Fig. 2). Re-evaluation of the parameter sets listed in Table 1 may improve function. Further, the introduction of a stochastic process to assign a value (e.g., between zero and the currently calculated upper value) according to a

statistical distribution (e.g., uniform or Gaussian) is another possibility. Regarding the quality of the carbon process, the evaluation of the decomposition rate $\kappa$, and relatedly, $\tau$ may be elaborated (Eqs. 1 and 3). Since one of the primary objectives of this study was to investigate quantitative impacts and functionality of allogenic (external; climatic or environmental conditioning) factors on long-term carbon dynamics, the current model does not specifically consider



autogenic (internal, ecosystem-dependent) aspects of the process. However, examination of the relative contribution of
allogenic and autogenic controls on carbon dynamics is important (Lund et al. 2010, van Bellen et al. 2011, Loisel and Yu
2013, Klein et al. 2013, Charman et al. 2015). Possible structure and processes to be incorporated include multiple pools of
different soil carbon stability (labile to recalcitrant) for inputs (i.e., litter fall) and outputs (i.e., rate of decomposition), and
their sensitivity to climate and/or hydrology (Boudreau and Ruddick 1991, Hilbert et al. 2000, Biasi et al. 2013).

The third problem relates to initial values for carbon and water to be specified for the Last Interglacial condition. Since we
had no prior information on initial values for that period, we started the integration with a uniform distribution for both soil
carbon and moisture at all grid points north of 50°N. We examined the model's sensitivity to initial values with a small set of
different SOC (namely, 5.0, 10.0, 20.0, 22.5, 25.0, 27.5, 30.0, 50.0, and 100.0 kgC m$^{-2}$) and soil moisture (similarly, 100,
500, 1000, 1500 mm for the 3000 mm column) values in limited locations (cf. Fig 5a). The model sensitivity showed clear
dependency on the initial values of SOC but was negligible for soil moisture. Based on this preliminarily examination, we
determined the initial values that would produce the most realistic range for the present-day circum-Arctic, namely, 25.0 kgC
m$^{-2}$ for SOC and 500 mm for soil moisture. This examination, however, was spatially limited to less than a dozen locations.
It is worth investigating the sensitivity to initial values with larger sets of locations, possibly with a nonuniform distribution
(i.e., starting with the present-day distribution under the Last Interglacial conditions).

**5. Conclusion**

In assessing and projecting the relative risks and impacts of permafrost degradation, the spatial distribution of SOC and ICE
provides essential information. However, uncertainties related to geographical distribution and the estimated range of the
total amount of stored carbon and ice obtained from synchronic compilations of samples or cores remain large. We adopted a
novel approach to estimate present-day spatial distribution and amounts through diachronic simulations. A conceptual
numerical model SOC-ICE-v1.0, representing the essential part of the cold-region subsurface carbon and water dynamics by
considering frozen ground (permafrost) and land cover changes (ice sheets and coastlines), was developed to calculate the
long-term balance of SOC and ICE. The model was integrated for a 125 thousand-year period from the Last Interglacial to
the present day for areas north of 50°N to simulate accumulations (or dissipations) of SOC and ICE in the circum-Arctic
region. Model performance was evaluated using observation-based data and evidence. Although the model was forced by
climate data constructed from a single Greenland ice core, the simulated time series reproduced temporal changes in northern
SOC and ICE at different climate locations well and successfully captured circum-Arctic regional differences in
characteristics. The model provided useful information for quantitative evaluation of the relative importance of allogenic
factors to control soil carbon dynamics under different climatological or topo-geographical conditions.



The set of simulated results can be compiled to produce snapshot maps of the geographical distributions of SOC and ICE in regions north of 50°N. One of these maps was used as the initial or boundary conditions in regional- to global-scale eco-climate models for future projections.

Despite its simplicity, the modelling framework employed in this study proved capable of accurately simulating evolution of

the circum-Arctic soil carbon and ground ice, and was powerful enough to provide their present-day spatial distributions. However, some improvements are required in the model, such as construction of more locally-specific forcing climate data series, improvement in the structure and parameterization of soil carbon dynamics in terms of inputs and outputs to the subsurface carbon pool, and determination of the initial value distributions for carbon and water integration.

**Code and data availability.**

The set of model codes, and sample driving and initial/boundary data to run the model are open for research purposes, and provided at https://github.com/MazaSaito/SOC-ICE/blob/master/SOC_ICE_sample.tar.gz (doi:10.5281/zenode.3839222), which also includes the user manual (https://github.com/MazaSaito/SOC-ICE/blob/master/User_manual.txt), and a sample shell code to run the model.

The open datasets used for determining the model parameters, and/or constructing the driving and boundary data are available from the respective data providers, i.e., SeaRISE (http://websrv.cs.umt.edu/isis/index.php/SeaRISE_Assessment), ICE-6G_C (https://www.atmosp.physics.utoronto.ca/~peltier/data.php), GLDAS (doi:10.5067/DLVU8VOPKN7L), CMIP5/PMIP3 (available from the ESGF nodes, e.g., https://esgf-node.ipsl.upmc.fr/projects/esgf-ipsl/), and reanalysis data (*UDel_AirT_Precip* at https://www.esrl.noaa.gov/psd/, and ERA-Interim at

https://www.ecmwf.int/en/forecasts/datasets/reanalysis-datasets/era-interim).

**Author contributions.**

KS proposed and managed the project, and designed both the study and the model. HM helped with model development and carried out numerical experiments. GI, HO, and TY helped in interpretation and validation of results. All authors read and approved the final manuscript.

**Competing interest**

The authors declare that they have no competing interests.



**Financial support.**

This study was conducted as a part of the Environment Research and Technology Development Fund project (2-1605) "Assessing and Projecting Greenhouse Gas Release from Large-scale Permafrost Degradation", supported by the Ministry of
Environment and the Environmental Restoration and Conservation Agency.

**Acknowledgments**

We thank Drs. Atsushi Sato and Hideki Miura for their advice. We also thank Dr. Jun'ichi Okuno for his comments and analysis in interpreting ICE-6G_C data. We are indebted to Drs. Akihiko Ito, Kazuhito Ichii, Hisashi Sato, and Hotaek Park for allowing use of GTMIP Stage 2 simulation data, and to Dr. Taku M. Saitoh for providing 'The compilation data set of
ecosystem functions in Asia (version 1.2)'. The *UDel_AirT_Precip* data were provided by the NOAA/OAR/ESRL PSD, Boulder, Colorado, USA, from their Web site at https://www.esrl.noaa.gov/psd/. The professional English language edit was performed by Editage, a division of Cactus Communications.

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
