# Peer review of "Conceptual Model to Simulate Long-term Soil Organic Carbon and Ground Ice Budget with Permafrost and Ice Sheets (SOC-ICE-v1.0)"

_Geoscientific Model Development, 2020_

## Referee Comment (RC1) · Anonymous Referee #1 · 28 Aug 2020

General comments:

The authors present a very timely and necessary modelling framework for assessing the spatial distribution of soil organic carbon (SOC) and ground ice (ICE) across the circumpolar permafrost region between the 50th and 70th latitudes. Moreover, the presented SOC-ICE-v1.0 model can be used to produce maps of these distributions at any time point during the last 125,000 years. This is obviously an ambitious task to initiate with, but the authors accomplish in providing modelling tools that have potential to inform about the history and future of permafrost-affected soils. In their recent manuscripts and published works the authors have already assessed future developments and published snapshot maps using outputs from SOC-ICE-v1.0.

Despite the simplified consideration of some relevant factors for SOC and ICE dynamics (very coarse representation of soil properties, only one ice core to force past circumpolar climate deviations), the models show promising performance in reconstructing SOC and ICE histories. What I find impressive is the model's ability to account for the role of continental ice sheets and changing sea level in the reconstructed time series for SOC and ICE. The manuscript is well written. Results are well presented and likely reproducible, although some of the performed pre-examinations are mentioned in a cursory manner.

Concerning the results, the time series over the last 125 ka appear mostly realistic, although the lack of validation data especially for ground-ice accumulation history hinders model evaluations. Despite the comparisons using observations from 8 locations across circumpolar north, I remain rather uninformed about the model's capability to reliably produce the actual spatial SOC and ICE variability. The authors state that the modelled SOC and ICE foremostly paint a picture of relative contents, that is, in relation to other grid cells across the study area and not absolute in situ contents. Rather coarse spatial analysis resolution and very coarse representation of soil properties additionally reduce the model's potential to address the local to regional consequences of organic carbon cycling to the atmospheric GHG or ground subsidence due to ground ice melt. Nevertheless, I consider that at the present SOC-ICE-v1.0 constitutes a fair step towards these goals.

I applaud the authors for their explicit explanations of the performed parameterisations. However, coming from a different modelling tradition, I have recognized and pointed out several places where I believe the methods would benefit from further clarification. Moreover, I have several specific comments and suggestions for the authors to consider before consideration of publication in GMD.

Specific comments:

lines 30-34: As ground ice is the other studied property, I would suggest adding a very brief note on what consequences its melt may have.

lines 39-40: While yedoma is a prominent type of ice-rich permafrost, all ice-rich permafrost is not exclusively yedoma but other types of ice-rich permafrost occur.

line 42: I would suggest avoiding the term "buried ice" in the context of ice wedges, as buried ice typically refers to ice accumulated on the ground surface (e.g. glacier, lake, river or sea ice) and later buried by sediments. See, e.g., Permafrost Subcommittee: 1988, Glossary of permafrost and related ground-ice terms, Associate Committee on Geotechnical Research, National Research Council of Canada, Ottawa, Technical Memorandum No. 142, 156 pp.

line 51: Please elaborate. What is "the aerial extent of ice-rich permafrost"?

I wonder if the whole introduction part would read more clearly if the descriptions of SOC accumulation history and research tradition (around the lines 71-84) would be embedded in section 1, and if section 1.1 would then solely focus on general descriptions of the model? Moreover, I am sure that the authors have become aware of a very recent study by Hugelius et al. (2020 PNAS), which appears to have provided notable advances in mapping the circumpolar C distribution. Consider updating parts of the review of current knowledge at lines 57-64 with the information provided therein. Hugelius et al. (2020) Large stocks of peatland carbon and nitrogen are vulnerable to permafrost thaw. PNAS 117 (34) 20438-20446

lines 90-93: The authors say that they incorporated a key parameter that represents temporal and spatial variations in climatic and topo-geographic conditions. This is related to the whole issue of external, or allogenic, factors, which are referred to in a bit inconsistent way by using terms, such as "climatic or environmental conditioning" (lines 517-518), "climatic, topographic and/or land composition" (399-400) or "climate, hydrology and topography" (77). I wonder if it would be possible to more explicitly describe what this parameter represents in this study. As far as I understand only continentality

(distance to the closest ocean body) and cover ratios of land, water, ice sheet and its thickness were specifically parameterized. DEM-based topographic conditions were only used in the authors' recently published paper (Saito et al. 2020 Progress in Earth and Planetary Science) to downscale the outputs of here presented model.

line 91: What curvatures? Terrain?

line 142: Can the authors very briefly clarify what they mean by stating that the Mosaic model outputs showed the best settings and results for regions north of 50°? No detailed explanations of the preliminary analyses are needed but please elaborate "settings and results".

lines 183-184: Does altitude data here refer to the thickness of an ice sheet or a digital elevation model? Related to this, clarifications on the used elevation data (if any) is needed. In the reference list the authors have Amante and Eakins (2009) and Tarboton (1989) related to DEM's but they are not cited in the text. Having read the authors' recent paper (Saito et al. 2020 Progress in Earth and Planetary Science) where they produced maps using SOC-ICE-v1.0, it appears that the related DEM was therein used to downscale model outputs.

Figure 3: Consider adding an explanation of the presented subsurface layers. Does the brown box refer to permafrost or impenetrable surface in general?

lines 295-297: Could this examination of "overall goodness of the reproduced time series" benefit from an elaboration or a reference?

Figure 4: The caption suggests that results from litter fall diagnosis are shown but they are missing. lines 315-318: Supplementary Fig. 2 could benefit from a more detailed caption (naming the 6 models, explaining the symbols).

Figure 5: Please align the panels to the same level and maybe label latitudes and longitudes.

lines 337-338, I had problems understanding this sentence. In the Discussion (lines

527-532), the authors provided a clear account on how the initial values for the spin-up were derived. I recommend presenting that piece of text in the Methods, so the spin-up is easier to understand. Please also consider elaborating what "5000 yr" means in this context – point in time or a period for which the model was spun up? Spin-up may also not be familiar for all readers, so maybe open that a little.

lines 368-369: Check language, some words seem to be missing from where the permafrost zone for Kevo site is mentioned.

line 371-372: Please revise the statement/language that Anaktavuk and Yakutsk locate "in areas that include the ice-rich permafrost (Yedoma) region". Yedoma, or other ice-rich permafrost regions, are not confined to these areas.

The results section has some sentences that would be better situated in the Discussion (e.g., lines 397-401, 407-411). Would it make sense to title this section Results and Discussion? The current Discussion is relatively short in comparison to the Results.

Chapter 3.2: In this chapter (more precisely, in section 3.2.3.), the authors also examine the simulated results of ice accumulation and dissipation, so it could be mentioned in this preamble (lines 430-436).

Chapter 3.2.1: I think that the authors do good job in discussing the possible reasons for the discrepancy between observed and simulated basal age. For example, the used climate data reconstruction from one ice core anticipatedly affects the results as the authors later discuss in 4.2. Related to this, in some point of the manuscript it would be beneficial to provide a brief reasoning behind using only one ice core and why it is suitable in the present purpose.

lines 472-473: I wonder about the large melting of ground ice during 14-15 ka, given that at least Kevo was under the continental ice sheet at that time. Is the anomalous melting related to glacial dynamics or warming climate, and also around 11 ka when the ice sheet finally retreated from the area? Could the authors say something more

precise about past glacial/ground-ice dynamics here in order to assess the reliability of the model as no independent observation-based validation data is available?

lines 501-502: Please revise the sentence. It could be made more readable, e.g. the expression "locality-prone profiles".

lines 527-532: Explaining this procedure would have seriously helped to understand the initial forcing values first presented in the beginning of the Results section. I thus suggest relocating this text to the Methods. Please also see my comment for lines 337-338.

line 536: I guess that by "relative" risks the authors here may refer to their earlier statement on how the model results do not necessarily represent the absolute SOC or ICE at a grid cell but rather their amounts relative to other grid cell? However, I think this is not clear in the first sentence of the Conclusion, and thus here "relative" could be removed.

lines 551-552: Do the authors here refer to another study using their model?

Technical corrections:

line 10: Is "relative" relevant or understandable without context?

line 39: "ice-rich-permafrost" to "ice-rich permafrost"

line 54: is "relative" needed?

line 58: Please correct "Gorham 19991"

line 114: In the abstract, the authors write "A conceptual and a numerical soil organic carbon–ground ice budget model". Are they separate models or one model as stated here ("The developed conceptual numerical model...")? Please be consistent through-out the text.

line 142: Rodell and Beaudoing ...., publication year missing

[Figure]

lines 175: "closest ocean, distance from the coast of the closest ocean" Is "closest ocean" redundant?

Table 1: kpice to kice

Table 2. Please consider explaining in the caption what the tau symbol denotes. Ta and Pr could also be explained. What does "Simulated ground ice is in meter" mean in footnote d?

line 394: Saito et al. 2020, not in review anymore.

line 419: Anaktavuku to Anaktavuk

line 420: length of the thawed layer?

lines 454-457: This information (starting from ", and then sorted to…") is found in the caption for Figure 8, and thus not necessary here.

At line 481, could the authors repeat the temporal resolution, i.e., for how long a period the snapshot maps can be compiled.

line 482: Saito et al. 2020 now published

line 491: There are Yokohata et al. 2020 a and b in the Refences, which one does this cite to? Is it published?

line 499: I think the last sentence; "Below is a list…" is not necessary here.

line 503: Thence to Hence

line 508: Do the referred timings of initiation refer to the results here, or by Morris et al. 2018?

line 513: "may improve function" could be clarified/said in a different way

line 532: I suggest editing; "less than a dozen" to "eight"

At least the following listed refs are not cited in the text:

- Amante and Eakins 2009

- AMAP 2017

- Biasi et al. 2005 (Biasi et al. 2013, however, is cited but not in the references)

- Bradley 1999

- Tarboton 1989

The following, in turn, not found in the References:

- Yu et al. 2008

- Brown et al. 1998

Please check all citations and references.

---

## Referee Comment (RC2) · Anonymous Referee #2 · 16 Sep 2020

General comments:

In the manuscript (MS), Saito et al. developed a numerical soil organic carbon–ground ice budget model (SOC-ICE-v1.0) to compute long-term evolution of soil organic carbon (SOC) and ground ice (ICE). The model was developed for the last 125 thousand years for areas north of 50°N. Based on the authors, the simulated results successfully (i) reproduced temporal changes in northern SOC and ICE, consistent with current knowledge and (ii) captured regional differences in different geographical and climatic characteristics within the circum-Arctic region. Moreover, the authors considered that the resulting circum-Arctic set of simulated time series can be compiled to produce

snapshot maps of SOC and ICE distributions for the past and present assessments or future projection simulations. Saito et al. concluded that the model provides substantial information on the temporal evolution and spatial distribution of circum-Arctic soil carbon and ground ice. However, model improvements in terms of, e.g., forcing climate data and choice of initial values are required in the future.

It is evident that the authors have addressed a topical issue, spatiotemporal prediction of soil organic carbon and ground ice across the circumpolar permafrost area. Moreover, the period of time is notable, the last 125 ka years. To my opinion, the topic of the MS fits well to Geoscientific Model Development (GMD). In general, I consider this MS to be relatively concise and well-written. However, I have two major concerns and some suggestions to improve the work.

First, there seems to be overlap between this MS and Saito et al. (2020) published in Progress in Earth and Planetary Science. Please clarify the novelty and added value of this MS. Second, how reliable are the results of SOC and ICE for areas covered by glaciers (e.g. continental ice sheets)? How these results relate to the fact that, for example, the site in northern Europe (Kevo) was covered by continental ice sheet until ca. 10 ka? The model seems to produce substantial variation in SOC despite the presence of glacier ice cover.

Saito, K., Machiya, H., Iwahana, G., Ohno, H., & Yokohata, T. (2020). Mapping simulated circum-Arctic organic carbon, ground ice, and vulnerability of ice-rich permafrost to degradation. Progress in Earth and Planetary Science, 7(1), 1-15.

Specific comments:

Title: Please reassess the use of 'conceptual' in the title. I would see the model as 'numerical' rather than 'conceptual'.

In the Abstract (and elsewhere), you use 'a conceptual and a numerical. . .'. For me a conceptual model differs from a numerical model but here the presented SOC-ICE-

v1.0 is both. Could you please clarify the motivation for the combination of conceptual and numerical?

It would be nice to have information on the spatial resolution of the model outputs somewhere in the Abstract. This could be relevant also in the Introduction or in the beginning of the section 2.

Introduction: maybe it would good to include definition of permafrost.

Lines 37-39: You state that '...well-recognized and widely examined using...' but refer only to one paper. Maybe few references more?

Lines 46-52: I would present the 'second pathway' and 'third pathway' in reverse order. The third is more significant pathway?

Line 59: Could Hugelius et al. (2020) published in PNAS be relevant here?

Hugelius, G., Loisel, J., Chadburn, S., Jackson, R. B., Jones, M., MacDonald, G., ... & Treat, C. (2020). Large stocks of peatland carbon and nitrogen are vulnerable to permafrost thaw. Proceedings of the National Academy of Sciences.

Lines 72-74: Need for so many references here?

Line 177: Why the warm period was set to start at 14 ka? For example, Holocene began ca. 11,5 ka before the present.

Sections 3.1, 3.2.1 and 3.2.2: I find it problematic to include references in the Result sections (results of this MS can be confused with published ones; look like discussion).

Lines 474-476: Do the literature support the mostly negative balance (accumulation rates) across the permafrost region for the past 12 ka? Please consider this in the Discussion.

Line 480 (also in the Abstract and Conclusion): You highlight the possibility to produce snapshot maps. Please provide some maps as examples in the MS.

Section 4.2: You focused on soil carbon in this section. How to improve the model outcomes related to ground ice?

Lines 551-552: The sentence ('One of these...') should be removed (not relevant here).

Table 1: In Eg. (4), why there are same figures for sand and clay? Their hydrological properties are different.

Table 2: There is no information for the 'Ta' and 'Pr' in the caption? If these area air temperature and average precipitation, please give information from what period they are? At least, some of the figures seem to be odd for modern annual averages.

Technical corrections:

Line 15: You could add 'permanently' (...permanently frozen...)

Line 18: You could add 'ground' (...and ground ice...)

Line (and elsewhere): Should the references be in chronological (or alphabetical) order?

Lines 58 and 64: Please correct Gorham 19991.

Line 107: Need to add 'soil' (soil carbon) and 'ground' (ground ice)?

Line 169: Should 'annual mean temperature' be 'MAAT'?

Lines 455-457: I would remove the sentence 'The lowest and highest whiskers of the box-whisker...'. This is good in caption but not needed here.

All abbreviations in the Figures and Tables should be spelled out in the captions.

---

## Author Comment (AC1) · 9 Nov 2020

Reply to the reviewers

"Conceptual Model to Simulate Long-term Soil Organic Carbon and Ground Ice Budget with Permafrost and Ice Sheets (SOC-ICE-v1.0)" by Kazuyuki Saito et al. in GMDD.

**Reviewer #1**

General comments:

The authors present a very timely and necessary modelling framework for assessing the spatial distribution of soil organic carbon (SOC) and ground ice (ICE) across the circumpolar permafrost region between the 50th and 70th latitudes. Moreover, the presented SOC-ICE-v1.0 model can be used to produce maps of these distributions at any time point during the last 125,000 years. This is obviously an ambitious task to initiate with, but the authors accomplish in providing modelling tools that have potential to inform about the history and future of permafrost-affected soils. In their recent manuscripts and published works the authors have already assessed future developments and published snapshot maps using outputs from SOC-ICE-v1.0.

Despite the simplified consideration of some relevant factors for SOC and ICE dynamics (very coarse representation of soil properties, only one ice core to force past circumpolar climate deviations), the models show promising performance in reconstructing SOC and ICE histories. What I find impressive is the model's ability to account for the role of continental ice sheets and changing sea level in the reconstructed time series for SOC and ICE. The manuscript is well written. Results are well presented and likely reproducible, although some of the performed pre-examinations are mentioned in a cursory manner.

Concerning the results, the time series over the last 125 ka appear mostly realistic, although the lack of validation data especially for ground-ice accumulation history hinders model evaluations. Despite the comparisons using observations from 8 locations across circumpolar north, I remain rather uninformed about the model's capability to reliably produce the actual spatial SOC and ICE variability. The authors state that the modelled SOC and ICE foremostly paint a picture of relative contents, that is, in relation to other grid cells across the study area and not absolute in situ contents. Rather coarse spatial analysis resolution and very coarse representation of soil properties additionally reduce the model's potential to address the local to regional consequences of organic carbon cycling to the atmospheric GHG or ground subsidence due to ground ice melt. Nevertheless, I consider that at the present SOC-ICE-v1.0 constitutes a fair step towards these goals.

I applaud the authors for their explicit explanations of the performed parameterisations. However, coming from a different modelling tradition, I have recognized and pointed out several places where I believe the methods would benefit from further clarification. Moreover, I have several specific comments and suggestions for the authors to consider before consideration of publication in GMD.

> We thank the reviewer for their sound and detailed review containing informative and constructive comments and suggestions. We have addressed each of these comments and suggestions in a point-to-point manner.
>
> Regarding the actual spatial SOC and ICE variability that the model can produce, we have explicitly demonstrated those reported by Saito et al. (2020, PEPS) for the present-day condition. Further, we have added Figure 10 containing sample snapshot maps for the LGM and mid-Holocene to the revised manuscript to

illustrate the spatial distribution and temporal variability under different climate conditions.

Specific comments:

lines 30-34: As ground ice is the other studied property, I would suggest adding a very brief note on what consequences its melt may have.

We added an explanation of the possible consequences of ground ice melt. (ll. 37–39)

lines 39-40: While yedoma is a prominent type of ice-rich permafrost, all ice-rich permafrost is not exclusively yedoma but other types of ice-rich permafrost occur.

We deleted the word "yedoma" to clarify the sentence. (ll. 45–47)

line 42: I would suggest avoiding the term "buried ice" in the context of ice wedges, as buried ice typically refers to ice accumulated on the ground surface (e.g. glacier, lake, river or sea ice) and later buried by sediments. See, e.g., Permafrost Subcommittee: 1988, Glossary of permafrost and related ground-ice terms, Associate Committee on Geotechnical Research, National Research Council of Canada, Ottawa, Technical Memorandum No. 142, 156 pp.

We deleted the word "buried" in response to the reviewer's comment, which we agree with. (l. 48)

line 51: Please elaborate. What is "the aerial extent of ice-rich permafrost"?

We modified the sentence to clarify it: "Although the spatial extent of the areas underlain by ice-rich permafrost with high soil carbon contents is limited, the impact of its degradation can reach wider areas globally." (ll. 57–59)

I wonder if the whole introduction part would read more clearly if the descriptions of SOC accumulation history and research tradition (around the lines 71-84) would be embedded in section 1, and if section 1.1 would then solely focus on general descriptions of the model? Moreover, I am sure that the authors have become aware of a very recent study by Hugelius et al. (2020 PNAS), which appears to have provided notable advances in mapping the circumpolar C distribution. Consider updating parts of the review of current knowledge at lines 57-64 with the information provided therein.

Hugelius et al. (2020) Large stocks of peatland carbon and nitrogen are vulnerable to permafrost thaw. PNAS 117 (34) 20438-20446

We thank the reviewer for providing new information. We moved the descriptions of SOC accumulation history and research tradition to section 1 and updated the description of the circumpolar C distribution mapping using the new information. (ll. 69–74)

lines 90-93: The authors say that they incorporated a key parameter that represents temporal and spatial variations in climatic and topo-geographic conditions. This is related to the whole issue of external, or allogenic, factors, which are referred to in a bit inconsistent way by using terms, such as "climatic or environmental conditioning" (lines 517-518), "climatic, topographic and/or land composition" (399-400) or "climate, hydrology and topography" (77). I wonder if it

would be possible to more explicitly describe what this parameter represents in this study. As far as I understand only continentality (distance to the closest ocean body) and cover ratios of land, water, ice sheet and its thickness were specifically parameterized. DEM-based topographic conditions were only used in the authors' recently published paper (Saito et al. 2020 Progress in Earth and Planetary Science) to downscale the outputs of here presented model.

In addition to direct climate control by temperature and precipitation, a key parameter $\tau$, the value of which reflects the climatic conditions (namely, freeze/thaw, a large value for frozen environment), soil condition (small values for coarse-grain soil), and topo-geographic condition (small values for steep, or well-drained areas) was introduced to represent temporal and spatial variations in climatic and topo-geographic conditions. The continentality (distance to the closest ocean body) and cover ratios of land, water, and ice sheet and its thickness are meant to constitute the boundary conditions for the simulations but are not parameters.

We realize that our explanation of the functionality of the key parameter, $\tau$, in section 2.2.2 (ll. 227–228 in the original manuscript) was not clear enough. We, thus, revised the following relevant sentences to clarify our intended meaning:

"We also incorporated a key parameter, $\tau$, that represented temporal and spatial variations in climatic and topo-geographic conditions (e.g. terrain curvatures, specific catchment areas, continentality, geomorphology, landscape, and fluvial conditions) to evaluate impacts on soil carbon evolution induced by these external factors, which we have discussed in section 2.2.2." (ll. 101–104)

"We also incorporated a key parameter, $\tau$, that represented temporal and spatial variations in climatic and topo-geographic conditions (e.g. terrain curvatures, specific catchment areas, continentality, geomorphology, landscape, and fluvial conditions) to evaluate impacts on soil carbon evolution induced by these external factors, which we have discussed in section 2.2.2." in section 1.1.

"This is the key parameter for the examination of climate and topo-geographic controls on soil carbon evolution, and we set the values of $\tau$ in this study in a geometric series at 4, 20, 100, and 500 yr, adhering to considerations" (section 2.2.2).

line 91: What curvatures? Terrain?

Yes, we meant terrain curvatures and have revised the term accordingly. (l. 102)

line 142: Can the authors very briefly clarify what they mean by stating that the Mosaic model outputs showed the best settings and results for regions north of 50_? No detailed explanations of the preliminary analyses are needed but please elaborate "settings and results".

We added a brief explanation to clarify our focus during the selection of model outputs: "After a preliminary analysis of the hydrological outputs of the four models (Noah, CLM, VIC, and Mosaic models), we selected the Mosaic model outputs because they yielded optimum results for cold regions north of 50 °N with respect to evaporation and runoff response to different soil types in the examined range of temperature, -30 $^{\circ}C$ to 15 $^{\circ}C$" (ll. 151–153)

lines 183-184: Does altitude data here refer to the thickness of an ice sheet or a digital elevation model? Related to this,

clarifications on the used elevation data (if any) is needed. In the reference list the authors have Amante and Eakins (2009) and Tarboton (1989) related to DEM's but they are not cited in the text. Having read the authors' recent paper (Saito et al. 2020 Progress in Earth and Planetary Science) where they produced maps using SOC-ICE-v1.0, it appears that the related DEM was therein used to downscale model outputs.

> The "altitude data" (we changed it to "orography condition" in line with the CMIP5/PMIP3 convention) refers to the elevation of the grid points used in each of the GCMs from the mean sea level. (l. 199)
>
> We deleted two citations, Amante and Eakins (2009) and Tarboton (1989), from the manuscript as these DEM-related data were used by Saito et al. (2020, PEPS) and not in this study as the reviewer correctly pointed out.

Figure 3: Consider adding an explanation of the presented subsurface layers. Does the brown box refer to permafrost or impenetrable surface in general?

> We added an explanation for the subsurface box, which, similar to a bucket-type model, does not resolve the vertical profile, but only retains the composition ratio.

lines 295-297: Could this examination of "overall goodness of the reproduced time series" benefit from an elaboration or a reference?

> We modified the sentence to clarify as "We examined the SOC time series calculated using the forcing data obtained from the CMIP5/PMIP3 models for the period (the Last Millennium and the historical runs), with respect to the mean values, temporal variability and smooth connectivity to the historical period, to select the IPSL simulation results." (ll. 306–308)

Figure 4: The caption suggests that results from litter fall diagnosis are shown but they are missing.

> As the litter fall diagnosis was illustrated in Figure 6 (left-most column), the associated words were deleted from this figure's caption.

lines 315-318: Supplementary Fig. 2 could benefit from a more detailed caption (naming the 6 models, explaining the symbols).

> We added an explanation for the 6 models and the symbol in Supplementary Fig. 2.

Figure 5: Please align the panels to the same level and maybe label latitudes and longitudes.

> We revised the panel for alignment and added the latitude and longitude labels.

lines 337-338, I had problems understanding this sentence. In the Discussion (lines 527-532), the authors provided a clear account on how the initial values for the spin-up were derived. I recommend presenting that piece of text in the Methods, so the spin-up is easier to understand.

> We reorganized the structure of the manuscript by moving both sentences that explain the initial values in the original "Discussion" section (ll. 527–532 in the original) along with the aforementioned sentence to a new

subsection, "2.3.3 Initial values and spin-up" to explain our selection of the initial values and the spin-up procedure used in this study.

Please also consider elaborating what "5000 yr" means in this context – point in time or a period for which the model was spun up? Spin-up may also not be familiar for all readers, so maybe open that a little.

We revised the sentence to clarify our intended meaning: "Before full integration, the model was spun up (or equilibrated) for a certain period of years to attain an internal balance in the SOC, soil moisture and ICE budget with the forcing. We integrated the model for 5000 years for spin-up with the constant forcing and perpetual boundary data taken from the 125 ka condition, starting from the uniform initial values of 25.0 kgC m$^{-2}$ of SOC and 500 mm of soil moisture at all grid points to reach an equilibrated state." (ll. 341–344)

lines 368-369: Check language, some words seem to be missing from where the permafrost zone for Kevo site is mentioned.

We revised the sentence for clarity. (ll. 356–358)

line 371-372: Please revise the statement/language that Anaktavuk and Yakutsk locate "in areas that include the ice-rich permafrost (Yedoma) region". Yedoma, or other ice-rich permafrost regions, are not confined to these areas.

We revised the statement as follows: "Anaktuvuk and Yakutsk are obtained from areas where the ice-rich permafrost can be found". (l. 360)

The results section has some sentences that would be better situated in the Discussion (e.g., lines 397-401, 407-411). Would it make sense to title this section Results and Discussion? The current Discussion is relatively short in comparison to the Results.

In adherence to the reviewers' advice, we merged sections 3 and 4 under "Results and discussions" section and divided section 3.1 to subsections 3.1.1 to 3.1.3 to improve organizational structure. Further, we revised the sentences throughout the section to clearly distinguish the results, cited values from the references, and discussions.

Chapter 3.2: In this chapter (more precisely, in section 3.2.3.), the authors also examine the simulated results of ice accumulation and dissipation, so it could be mentioned in this preamble (lines 430-436).

We mentioned ground ice simulations as the reviewer suggested. (ll. 445-447)

Chapter 3.2.1: I think that the authors do good job in discussing the possible reasons for the discrepancy between observed and simulated basal age. For example, the used climate data reconstruction from one ice core anticipatedly affects the results as the authors later discuss in 4.2. Related to this, in some point of the manuscript it would be beneficial to provide a brief reasoning behind using only one ice core and why it is suitable in the present purpose.

We revised a paragraph in section 3.3.2 (former 4.2) to state the reasons for using only single-core reconstruction in this study. (ll. 528–534)

lines 472-473: I wonder about the large melting of ground ice during 14-15 ka, given that at least Kevo was under the continental ice sheet at that time. Is the anomalous melting related to glacial dynamics or warming climate, and also around 11 ka when the ice sheet finally retreated from the area? Could the authors say something more precise about past glacial/ground-ice dynamics here in order to assess the reliability of the model as no independent observation-based validation data is available?

> We argue that this apparent synchronous melting in relatively southern areas (i.e. compared to high-latitude Alaska or west Canada) occurred because the reconstructed temperature with the warming peak of the SeaRISE time series (likely for the Bølling-Allerød interstadial) exceeded the melting threshold for those regions. We added the following information regarding the simulated extensive melting of ground ice.
>
> "These apparent synchronous melting in these areas likely resulted from the single-sourced warming peak of the Bølling-Allerød interstadial in the SeaRISE time series (e.g. Fig. 4a-c) and suggests the need of further studies to include local climate variations to the driving data". (ll. 488–490)

lines 501-502: Please revise the sentence. It could be made more readable, e.g. the expression "locality-prone profiles".

> We revised the sentence as follows: "…forcing data should be designed to accommodate information that is more specific to the local history so that the resulting time series and maps can reflect regional diversity and characteristics more adequately". (ll. 543–544)

lines 527-532: Explaining this procedure would have seriously helped to understand the initial forcing values first presented in the beginning of the Results section. I thus suggest relocating this text to the Methods. Please also see my comment for lines 337-338.

> We moved these sentences to the new subsection "2.3.3 Initial values and spin-up" and added an explanation for the spin-up procedure.

line 536: I guess that by "relative" risks the authors here may refer to their earlier statement on how the model results do not necessarily represent the absolute SOC or ICE at a grid cell but rather their amounts relative to other grid cell? However, I think this is not clear in the first sentence of the Conclusion, and thus here "relative" could be removed.

> The overall study objective was to assess the relative contributions of the three pathways to greenhouse gas release by permafrost degradation. However, as it has little relevance in regard to the scope of this manuscript, we deleted the term "relative" throughout the text to improve clarity. (l. 577)

lines 551-552: Do the authors here refer to another study using their model?

> We deleted the sentence as it was not relevant to the context.

Technical corrections:

line 10: Is "relative" relevant or understandable without context?

> The overall study objective was to assess the relative contributions from the three pathways to greenhouse gas

release due to permafrost degradation. However, since it is less relevant concerning the scope of this manuscript, we deleted the term "relative" throughout the text to improve clarity. (l. 11)

line 39: "ice-rich-permafrost" to "ice-rich permafrost"

We have corrected it. (l. 45)

line 54: is "relative" needed?

We deleted this term throughout the text to improve clarity. (l. 61)

line 58: Please correct "Gorham 19991"

We corrected to Gorham (1991). (l. 65, l. 73)

line 114: In the abstract, the authors write "A conceptual and a numerical soil organic carbon–ground ice budget model". Are they separate models or one model as stated here ("The developed conceptual numerical model...")? Please be consistent throughout the text.

We deleted the term "conceptual" from the title, abstract, and main text to clarify that the model proposed in the paper is a numerical model.

line 142: Rodell and Beaudoing ...., publication year missing.

We added the publication year (2007). (l. 154)

lines 175: "closest ocean, distance from the coast of the closest ocean" Is "closest ocean" redundant?

We removed the first occurrence of "closest ocean" to reduce redundancy. (l. 189)

Table 1: kpice to kice

We have corrected it accordingly.

Table 2. Please consider explaining in the caption what the tau symbol denotes. Ta and Pr could also be explained. What does "Simulated ground ice is in meter" mean in footnote d?

We added an explanation for tau symbol, as well as for the abbreviation Ta and Pr, in the caption and in the footnote. We revised the explanation in the footnote d) for clarity.

line 394: Saito et al. 2020, not in review anymore.

We updated the citation. (l. 409)

line 419: Anaktavuku to Anaktavuk

We checked and corrected the spelling in texts, figures, and tables, following the notations in Iwahana et al. (2016). (l. 360)

line 420: length of the thawed layer?

We changed to "length of the thawing period and depth of the thawed layer". (l. 435)

lines 454-457: This information (starting from ", and then sorted to...") is found in the caption for Figure 8, and thus not necessary here.

We deleted the sentences accordingly.

At line 481, could the authors repeat the temporal resolution, i.e., for how long a period the snapshot maps can be compiled.

We mentioned that the temporal resolution is annual (l. 498). Theoretically (i.e. without practical concerns and limitations on storage size), any year during the 125 kyr integration can be compiled to make a snapshot map.

line 482: Saito et al. 2020 now published

We updated the citation. (l. 510)

line 491: There are Yokohata et al. 2020 a and b in the Refences, which one does this cite to? Is it published?

As Yokohata et al. (2020a) is less relevant to our study, we deleted it, and revised Yokohata et al. (2020b) to Yokohata et al. (2020). (l. 518)

line 499: I think the last sentence; "Below is a list..." is not necessary here.

We have deleted the sentence.

line 503: Thence to Hence

We revised the relevant sentences. (ll. 535–537)

line 508: Do the referred timings of initiation refer to the results here, or by Morris et al. 2018?

We revised the sentence to indicate that they were reported by Morris et al. (2018). (ll. 540–542)

line 513: "may improve function" could be clarified/said in a different way

We revised the sentence for clarity. (ll. 548-550)

line 532: I suggest editing; "less than a dozen" to "eight"

We have corrected to "eight". (l. 573)

At least the following listed refs are not cited in the text:

- Amante and Eakins 2009: removed from the list
- AMAP (SWIPA) 2017: cited in the text (l. 32, l. 39)

- Biasi et al. 2005 (Biasi et al. 2013, however, is cited but not in the references): Biasi et al. (2013) replaced with Biasi et al. (2005) in the text (l. 559-560)

- Bradley 1999: removed from the list

- Tarboton 1989: removed from the list

The following, in turn, not found in the References:

- Yu et al. 2008: corrected to Yu et al. 2009 in the text (l. 468)

- Brown et al. 1998: added in the list

Please check all citations and references. We checked all the citations and references.

**Reviewer #2:**

General comments:

In the manuscript (MS), Saito et al. developed a numerical soil organic carbon–ground ice budget model (SOC-ICE-v1.0) to compute long-term evolution of soil organic carbon (SOC) and ground ice (ICE). The model was developed for the last 125 thousand years for areas north of 50_N. Based on the authors, the simulated results successfully (i) reproduced temporal changes in northern SOC and ICE, consistent with current knowledge and (ii) captured regional differences in different geographical and climatic characteristics within the circum-Arctic region. Moreover, the authors considered that the resulting circum-Arctic set of simulated time series can be compiled to produce snapshot maps of SOC and ICE distributions for the past and present assessments or future projection simulations. Saito et al. concluded that the model provides substantial information on the temporal evolution and spatial distribution of circum-Arctic soil carbon and ground ice. However, model improvements in terms of, e.g., forcing climate data and choice of initial values are required in the future.

It is evident that the authors have addressed a topical issue, spatiotemporal prediction of soil organic carbon and ground ice across the circumpolar permafrost area. Moreover, the period of time is notable, the last 125 ka years. To my opinion, the topic of the MS fits well to Geoscientific Model Development (GMD). In general, I consider this MS to be relatively concise and well-written. However, I have two major concerns and some suggestions to improve the work.

> We thank the reviewer for the appreciation, and for the questions and comments that helped us in enhancing the scientific as well as expressional contents of the manuscript. We have provided a point-to-point reply to the comments and suggestions below:

First, there seems to be overlap between this MS and Saito et al. (2020) published in Progress in Earth and Planetary Science. Please clarify the novelty and added value of this MS.

> Saito, K., Machiya, H., Iwahana, G., Ohno, H., & Yokohata, T. (2020). Mapping simulated circum-Arctic organic carbon, ground ice, and vulnerability of ice-rich permafrost to degradation. Progress in Earth and Planetary Science, 7(1), 1-15.

> This manuscript aims to describe the detailed constructions of the novel numerical model (Sections 2.2 and 2.3) along with the evaluations of the simulated results (time series) in terms of the spatial variations (Section 3.1) and dynamic behaviours (Section 3.2) in SOC, ICE and soil moisture.

> In contrast, Saito et al. (2020, PEPS) aimed to demonstrate the applicability of the simulated results for a specific time, namely the present-day, and downscaled the results with other topographical and hydrological information to produce spatial maps of SOC and ICE with relatively finer horizontal resolution and to evaluate the vulnerability distribution of ice-rich permafrost for the degradation. We agree that these originally-intended differences were weakened during the process of reviewing the PEPS manuscript as we were requested by a reviewer to include a relatively detailed description of the model construct and evaluations on the time series of the forcing and simulated data.

Second, how reliable are the results of SOC and ICE for areas covered by glaciers (e.g. continental ice sheets)? How these results relate to the fact that, for example, the site in northern Europe (Kevo) was covered by continental ice sheet

until ca. 10 ka? The model seems to produce substantial variation in SOC despite the presence of glacier ice cover.

> The value of SOC is initialized (i.e. made to null) and ICE is allowed to change when the overlying ice sheet retreats, as described in section 2.3.2. In the current model, the extents of changes in ice sheets (coverage or retreat) are determined by the ICE-6G_C dataset at the 1° horizontal resolution. Sub-grid-scale changes (e.g. extent of ice sheet shrinking, changes in coastline locations, and submergence/uplift of land within the 1° grid) are only considered through changes in occupancy proportion. As shown in Fig. 6 (second from the right, 6[th] row), the ice sheet (in yellow) dominantly covered the grid point closest to Kevo until approximately 10 ka; the ground ice started melting and soil water level fluctuated from 10 ka. We agree that the appropriateness of allowing for the accumulation of ground ice under the ice sheet cover condition, as discussed in section 3.1.2 (formerly 3.1), requires further consideration. We appreciate the reviewer for bringing this to our attention.

Specific comments:

Title: Please reassess the use of 'conceptual' in the title. I would see the model as 'numerical' rather than 'conceptual'. In the Abstract (and elsewhere), you use 'a conceptual and a numerical...'. For me a conceptual model differs from a numerical model but here the presented SOC-ICE v1.0 is both. Could you please clarify the motivation for the combination of conceptual and numerical?

> Originally, we aimed to indicate that this study is a numerical realization of a conceptual two-box model. However, as both reviewers suggest, it is very confusing. Thus, we removed the word "conceptual" to improve clarity.

It would be nice to have information on the spatial resolution of the model outputs somewhere in the Abstract. This could be relevant also in the Introduction or in the beginning of the section 2.

> We added information on the spatial resolution (i.e. 1°) of the simulations. (l. 25, l. 93, and l. 126)

Introduction: maybe it would good to include definition of permafrost.

> We obtained the definition of permafrost from the IPA's glossary definition and added it to the introduction (van Everdingen, 1998). (ll. 32–34)

Lines 37-39: You state that '...well-recognized and widely examined using...' but refer only to one paper. Maybe few references more?

> We revised the sentence to "…well-recognized and widely examined using global-scale models including Earth System Models (ESMs) and Global Climate Models (GCMs)", and added three more references. (ll. 42–45)

Lines 46-52: I would present the 'second pathway' and 'third pathway' in reverse order. The third is more significant pathway?

> We agree that the third pathway (or secondary release by decomposition of soil carbon newly exposed by permafrost degradation) is significant than the second pathway (direct release of bubbles trapped in ice). However, the direct release likely occurs earlier than the secondary one. Thus, we retained the original order.

Line 59: Could Hugelius et al. (2020) published in PNAS be relevant here?

Hugelius, G., Loisel, J., Chadburn, S., Jackson, R. B., Jones, M., MacDonald, G., ... & Treat, C. (2020). Large stocks of peatland carbon and nitrogen are vulnerable to permafrost thaw. Proceedings of the National Academy of Sciences.

Thank you for providing this information; we added the citation to the manuscript and revised the paragraph. (ll. 69–74)

Lines 72-74: Need for so many references here?

We selected the references according to their relevance and significance to the context of the study. (ll. 76–77)

Line 177: Why the warm period was set to start at 14 ka? For example, Holocene began ca. 11,5 ka before the present.

We set to start the warm period at 14 ka to include the warm environment under the Bølling-Allerød interstadial and avoided quick reversals afterwards via the Younger Dryas and so on. Observation of the sensitivity of the background climatic conditions will be problematic.

Sections 3.1, 3.2.1 and 3.2.2: I find it problematic to include references in the Result sections (results of this MS can be confused with published ones; look like discussion).

We reorganized the structure of the manuscript (sections 3 and 4 are now combined under one section: "Results and discussions") and revised the sentences to make distinctions between the results of this study, those from related studies, and discussions.

Lines 474-476: Do the literature support the mostly negative balance (accumulation rates) across the permafrost region for the past 12 ka? Please consider this in the Discussion.

We revised this paragraph and elaborated the discussion using reported literature from relevant studies. (ll. 475–480)

Line 480 (also in the Abstract and Conclusion): You highlight the possibility to produce snapshot maps. Please provide some maps as examples in the MS.

We provided sample snapshot maps for the LGM and mid-Holocene with different $\tau$ values in Figure 10. Another example of the present-day (year 0 = 1950) is already presented in Figure 5a–d with reference to Saito et al. (2020) (ll. 499–509)

Section 4.2: You focused on soil carbon in this section. How to improve the model outcomes related to ground ice?

We added text on future improvement with respect to hydrology and ice dynamics in this paragraph. (ll. 561–569)

Lines 551-552: The sentence ('One of these...') should be removed (not relevant here).

We agree and have deleted the sentence.

Table 1: In Eg. (4), why there are same figures for sand and clay? Their hydrological properties are different.

It was a typo; the value for clay is 0.03.

Table 2: There is no information for the 'Ta' and 'Pr' in the caption? If these area air temperature and average precipitation, please give information from what period they are? At least, some of the figures seem to be odd for modern annual averages.

We added an explanation for the abbreviation Ta and Pr and the period over which climatology was calculated in the caption and footnote. We checked and corrected some figures for Ta and Pr. However, site names such as Fairbanks and Kevo are labels representing the nearest 1° resolution grid point, at which the Ta and Pr values are obtained in the aggregated climate dataset. In some cases, they may be different. We revised the explanation in the caption to improve clarity.

Technical corrections:

Line 15: You could add 'permanently' (...permanently frozen...)

We added this word to the text. (l. 14)

Line 18: You could add 'ground' (...and ground ice...)

We revised to use the abbreviations (i.e. SOC and ICE) in the abstract.

Line (and elsewhere): Should the references be in chronological (or alphabetical) order?

We corrected the references in chronological order throughout the text.

Lines 58 and 64: Please correct Gorham 19991.

We revised this to Gorham (1991). (l. 65, l. 73)

Line 107: Need to add 'soil' (soil carbon) and 'ground' (ground ice)?

We added the words "soil" (soil carbon) and "ground" (ground ice). (l. 117)

Line 169: Should 'annual mean temperature' be 'MAAT'?

We revised this to "MAAT". (l. 183)

Lines 455-457: I would remove the sentence 'The lowest and highest whiskers of the box-whisker...'. This is good in caption but not needed here.

We deleted this sentence.

All abbreviations in the Figures and Tables should be spelled out in the captions.

We spelt all the abbreviations in the Figures and Tables.

---

## Author Response (AR2)

Reply to the Reviewer

I thank the authors for taking into account all of my comments on their manuscript. I think that changes to the overall structure of the manuscript, as well as the clarifications of the methods in many places have made it much easier to read and acceptable to be published after some technical corrections.

We thank again for the reviewer for their thorough inspection of our revision. We have provided a point-to-point reply to the considerate comments and suggestions below:

ll. 197-199 Please check language, is "of" needed after because?

We have deleted the word.

Figure 4 caption: No need to mention the litter fall time series here as they are not included in the figure.

We have deleted the sentence.

l. 530 "relatively more time" ? Please, check language.

We have revised the entire sentence. Please see our response to the next point.

ll. 528-533: I am happy that the authors went their way to reason why only a single ice core dataset was used to reconstruct climate histories. However, I am not entirely convinced by the first point (1) which basically states that more ice core data were not included because they would have taken more time compile and/or they could have contradicted with the one that was used.

We agree and have revised the first point (1) to "it enabled to minimize the degree of subjectivity and uncertainty injected when compiled from different, sometimes contradicting and/or fragmentary, ice core data" (ll. 530–531).

There are many cases where references are not in chronological order, which, I understand, is the authors' citing style of choice.

We reexamined the order of references, and have changed entirely to chronological order for clarity and concord.

[revised manuscript text omitted]